# SCOUT: Cyclic Causal Discovery Under Soft Interventions with Unknown Targets

**Alpar Turkoglu** [1]   **Muralikrishnna G. Sethuraman** [1]   **Faramarz Fekri** [1]

## Abstract

Learning causal relationships between variables from data is a fundamental research area with many applications across disciplines. Most of the existing causal discovery algorithms rely on the assumptions that (i) the underlying system is acyclic, (ii) the exogenous noise variables are Gaussian, and (iii) that the intervention targets for the data generating experiments are known. While these assumptions simplify the analysis, they are violated in real-life systems. Most existing methods that address these issues either assume the underlying model is linear or are constrained to operate in limited interventional settings. To that end, we propose SCOUT, a novel causal discovery framework to learn nonlinear causal cyclic relationships from soft interventional data with unknown targets. Our main approach maximizes the data log-likelihood to recover the graph structure, using two normalizing-flow architectures—contractive residual flows and neural spline flows. By conducting experiments on synthetic and real-world data, we show that SCOUT outperforms state-of-the-art methods in both causal graph and unknown target recovery across various interventional and noise settings.

## 1. Introduction

Identifying cause-effect relations among variables is a fundamental challenge across many scientific fields. Causal models provide a mechanistic understanding of underlying systems, enabling us to predict how they behave under previously unseen perturbations. Typically, causal interactions are encoded as a directed graph (DG), reducing the problem

---
[1]School of Electrical and Computer Engineering, Georgia Institute of Technology. Correspondence to: Alpar Turkoglu <aturkoglu3@gatech.edu>, Muralikrishnna G. Sethuraman <muralikgs@gatech.edu>.

*Proceedings of the 43rd International Conference on Machine Learning*, Seoul, South Korea. PMLR 306, 2026. Copyright 2026 by the author(s).

of recovering causal effects to identifying the structure of this graph.

Causal discovery methods can be broadly categorized into three classes: (i) constraint-based, (ii) score-based, and (iii) hybrid methods. *Constraint-based* methods, such as the PC algorithm (Spirtes et al., 2001; Triantafillou & Tsamardinos, 2015; Heinze-Deml et al., 2018), exploit the conditional independence relations implied by the underlying causal graph and aim to recover a graph that is consistent with the observed independencies. These methods typically suffer from scalability issues, as they require testing a large number of conditional independence relations, which grows exponentially with the number of variables in the graph.

*Score-based* methods, such as GES (Meek, 1997; Hauser & Bühlmann, 2012), instead formulate causal discovery as an optimization problem, seeking to maximize a penalized score function—such as the Bayesian Information Criterion (BIC)—over the space of candidate graphs. Since the number of possible graphs grows super-exponentially with the number of nodes, these methods generally rely on greedy or heuristic search strategies to remain computationally tractable. A major recent breakthrough was introduced by Zheng et al. (2018), who proposed a smooth characterization of the acyclicity constraint, enabling optimization over continuous adjacency matrices while restricting the solution to directed acyclic graphs (DAGs). This idea has inspired numerous extensions (Yu et al., 2019; Ng et al., 2020; Ng et al.; Zheng et al., 2020; Lee et al., 2019; Brouillard et al., 2020) that cast causal discovery as a continuous optimization problem under various modeling assumptions and constraints. Finally, *hybrid methods* (Tsamardinos et al., 2006; Solus et al., 2017; Wang et al., 2017) combine elements of both constraint-based and score-based approaches, typically by incorporating conditional independence information into a scoring framework or by using independence tests to guide and prune the search over graph structures.

With a few notable exceptions (Hyttinen et al., 2012; Richardson, 1996; Mooij & Heskes, 2013; Bongers et al., 2021), most existing causal discovery methods assume that the underlying causal graph is acyclic and operate purely in the observational regime. While these assumptions substantially simplify the search space and facilitate theoret-

ical analysis, they are often unrealistic in real-world systems, where feedback mechanisms are common (Sachs et al., 2005; Freimer et al., 2022). Moreover, enforcing acyclicity typically increases the computational complexity of both optimization- and search-based procedures.

Recent advances in experimental sciences, particularly in biology, have enabled the collection of large-scale interventional datasets. For instance, technological developments in biological assays building upon CRISPR/Cas9 and single-cell RNA sequencing (Dixit et al., 2016) now make it possible to probe a large number of interventions in gene regulatory networks. This, in turn, creates a growing need for causal discovery algorithms that can effectively leverage multiple and heterogeneous interventional contexts. However, even recent works on cyclic causal discovery fall short of full generality, as they typically rely on restrictive assumptions such as linearity (Hyttinen et al., 2012; Rothenhäusler et al., 2015) or specific noise models (e.g., Gaussianity), and often consider only surgical (hard) interventions (Sethuraman et al., 2023; Sethuraman & Fekri, 2025). These limitations substantially restrict their applicability to complex real-world systems.

To address these challenges, we propose a novel causal discovery framework, SCOUT, that simultaneously accommodates *nonlinear mechanisms, directed cycles, non-Gaussian additive noise, and soft interventional experiments with unknown targets*, thereby substantially broadening the scope of causal discovery in realistic experimental settings.

## 1.1. Related Works

**Cyclic Graphs.** A range of methods have been proposed to address feedback loops in causal graphs. Early work by Richardson (1996) extended constraint-based approaches from DAGs to graphs with directed cycles, while Lacerda et al. (2008) generalized ICA-based causal discovery to linear cyclic models under non-Gaussian noise. More recent efforts have focused on score-based formulations for learning cyclic structures (Huetter & Rigollet, 2020; Amendola et al., 2020; Mooij & Heskes, 2013; Drton et al., 2019), with some methods further leveraging interventional data to improve identifiability (Hyttinen et al., 2012; Huetter & Rigollet, 2020). Building on these ideas, Sethuraman et al. (2023) proposed a differentiable, likelihood-based framework for learning nonlinear cyclic graphs that avoids explicit acyclicity constraints by directly optimizing the data likelihood. Subsequent extensions broaden this framework to account for unmeasured (latent) variables (Sethuraman & Fekri, 2025).

**Soft interventions.** There are various approaches for causal discovery under soft interventions. Greedy Interventional Equivalence Search (GIES) (Hauser & Bühlmann, 2012) extends greedy equivalence search for DAGs to interventional data. It is a score-based algorithm that operates under known multiple intervention targets for each experiment. IGSP (Wang et al., 2017) instead uses permutation-based causal inference, which is non-parametric, so it doesn't rely on the Gaussian assumption. The Joint causal inference framework (JCI) (Mooij et al., 2016) extends the ideas of classical constraint-based algorithms for interventions by adding context variables to the graph. Differentiable Causal Discovery from Interventions (DCDI) (Brouillard et al., 2020) uses a continuous-optimization approach in the interventional setting to learn the ground-truth DAG by maximizing the likelihood across all datasets. Backshift (Rothenhäusler et al., 2015) is an algorithm for linear causal cyclic models that uses different shift interventions for structure learning. Kocaoglu et al. (2019) characterize causal graph equivalence under soft interventions in the presence of latent variables.

**Unknown targets.** Several methods have been proposed in the literature to estimate the graph structure of a causal system when the intervention targets are unknown. UT-IGSP (Squires et al., 2020) is a version of IGSP that has been extended to work with unknown targets. More recent methods developed by Varici et al. (2022) and Yang et al. (2024), detect unknown targets by exploiting sparse changes in the precision matrices or in the noise distributions, respectively. Bayesian Causal Discovery with Unknown Interventions (BaCaDI) (Hägele et al., 2022) uses a variational inference approach to learn the DAG structure and the intervention targets. JCI (Mooij et al., 2016), DCDI (Brouillard et al., 2020), and BackShift (Rothenhäusler et al., 2015) also estimate the unknown intervention targets while recovering the causal graph structure. Finally, Jaber et al. (2020) study causal discovery under soft interventions with unknown targets while allowing for latent confounding.

## 1.2. Contribution

In this work, we address four major challenges in causal discovery: *directed cycles*, *nonlinearity*, *non-Gaussian exogenous noise*, and *soft interventions with unknown targets*. Our main contributions can be summarized as follows:

- We propose SCOUT, a novel framework for causal discovery that utilizes the normalizing flows architectures of contractive residual flow and neural spline flows to learn nonlinear cyclic causal relationships under non-Gaussian noise from soft interventional data with unknown targets, while simultaneously inferring the intervention targets.

- We prove that exact maximization of the proposed score function identifies the interventional equivalence class of the ground-truth graph.

• We perform extensive experiments, benchmarking SCOUT against state-of-the-art causal discovery methods on both synthetic and real-world datasets.

### 1.3. Organization

The remainder of the paper is organized as follows: Section 2 describes the problem setup. Section 3 introduces SCOUT, our framework for nonlinear cyclic causal discovery under soft interventions with unknown targets. Section 4 presents experimental results on synthetic and real-world datasets. Finally, we conclude the paper with Section 5.

## 2. Problem Setup

### 2.1. Structural Equations for Cyclic Causal Graphs

Let $\mathcal{G} = (V, E)$ be a cyclic causal graph, where $V$ denotes the vertex set $\{1, \ldots, d\}$ and $E$ denotes the directed edges of the form $i \to j$. Each node $i \in V$ is associated with a random variable $X_i$, and each edge $i \to j \in E$ represents a direct causal relation from random variable $X_i$ to $X_j$. Following the framework proposed by Bollen (1989) and Pearl (2009), we use the structural equation model (SEM) to represent our system, that is:

$$X_i = f_i(\boldsymbol{X}_{pa_{\mathcal{G}}(i)}) + \epsilon_i, \quad i = 1, \ldots, d, \qquad (1)$$

where $pa_{\mathcal{G}}(i) := \{j \in V : j \to i \in E \text{ and } j \neq i\}$ denotes the *parent set* of $X_i$ in $\mathcal{G}$. $\boldsymbol{X}_{pa_{\mathcal{G}}(i)}$ is the random vector consisting of the collection of these parents. The function $f_i$ represents the *causal mechanism* encoding the functional relationships between the random variable $X_i$ and its parents. $\epsilon_i$ is the *exogenous noise* variable accounting for the stochastic nature of our system and can be non-Gaussian in our model. Note that we exclude self-loops in this model to avoid dealing with extra identifiability issues (Bongers et al., 2021; Hyttinen et al., 2012).

For convenience, (1) can be combined over all nodes $i \in \{1, \ldots, d\}$ into a vectorized form by collecting the causal mechanism into a joint function $\boldsymbol{f} = (f_1, \ldots, f_d)$ in the following way:

$$\boldsymbol{X} = \boldsymbol{f}(\boldsymbol{X}) + \boldsymbol{\epsilon}. \qquad (2)$$

Note that the SEM in (2) induces a probability distribution over the exogenous noise variables $p_E(\epsilon)$. We assume the system is free of confounders (causal sufficiency), as a result the exogenous noise variables are independent of each other.

Due to the (potential) presence of cycles in the SEM, the observations $\boldsymbol{X}$ can be thought of as a snapshot of a dynamical process under equilibrium conditions. For a random draw of $\boldsymbol{\epsilon}$, $\boldsymbol{X}$ is the solution to (2). We also assume that there is a fixed, unique solution for each draw of $\boldsymbol{\epsilon}$, which allows us to define an invertible *forward map* $\boldsymbol{X} \mapsto \boldsymbol{\epsilon} = (\mathbf{id} - \boldsymbol{f})(\boldsymbol{X})$, where $\mathbf{id}$ represents the identity transformation. A more

detailed discussion of solvability for cyclic systems under equilibrium can be found in Appendix A.

Under these assumptions, the probability density function for $\boldsymbol{X}$ is well-defined and can be written as:

$$p_{\mathcal{G}}(\mathbf{X}) = p_E((\mathbf{id} - \boldsymbol{f})(\boldsymbol{X})) \left| \det(\mathbf{J}_{(\mathbf{id}-\boldsymbol{f})}(\boldsymbol{X})) \right|, \quad (3)$$

where $\mathbf{J}_{(\mathbf{id}-\boldsymbol{f})}(\boldsymbol{X})$ denotes the Jacobian matrix of the function $(\mathbf{id} - \boldsymbol{f})$ evaluated at $\boldsymbol{X}$.

### 2.2. Modeling Interventions

One important feature of causal graphs is that they can be used to infer the model's behavior under interventions. In this work, we focus on imperfect interventions, also known as soft interventions. In contrast to surgical (hard) interventions, the connectivity of the intervened node with its parents is preserved under these types of interventions. Still, the causal mechanism or noise characteristics may be altered depending on the type of soft intervention.

Given a set of of intervened nodes $\mathcal{I} \subseteq V$ the SEM in (1) takes the following form:

$$X_i = \begin{cases} \tilde{f}_i^{(k)}(\boldsymbol{X}_{pa_{\mathcal{G}}(i)}) + \tilde{\epsilon}_i^{(k)}, & \text{if } i \in I_k, \\ f_i(\boldsymbol{X}_{pa_{\mathcal{G}}(i)}) + \epsilon_i, & \text{if } i \notin I_k, \end{cases} \qquad (4)$$

where $\tilde{f}_i^{(k)}$, $\tilde{\epsilon}_i^{(k)}$ denote the intervened causal mechanism and intervened noise variable for the k-th experiment respectively.

We consider $K$ interventional experiments where $I_k \in \mathcal{I}$ represent the interventional targets for the $k$-th experiment. Similar to the observational setting, we can combine (4) over all the nodes $i \in V$ to obtain the following vectorized form:

$$\boldsymbol{X} = \mathbf{U}_k(\boldsymbol{f}(\boldsymbol{X}) + \boldsymbol{\epsilon}) + (\mathbf{I}_d - \mathbf{U}_k)(\tilde{\boldsymbol{f}}(\boldsymbol{X}) + \tilde{\boldsymbol{\epsilon}}), \quad (5)$$

where $\mathbf{U}_k \in \{0, 1\}^{d \times d}$ is a diagonal matrix indicating which variables are observed in the $k$-th experiment, in other words, $(\mathbf{U}_k)_{ii} = 1$ if $i \notin I_k$ and $(\mathbf{U}_k)_{ii} = 0$ if $i \in I_k$. $\mathbf{I}_d$ is the $d \times d$ identity matrix. In this work, we treat $\mathbf{U}_k$ as an unknown parameter to be learned during training, given the distinct experiment indices $k$.

For each interventional experiment, the forward map $\boldsymbol{X} \mapsto \mathbf{U}_k\boldsymbol{\epsilon} + (\mathbf{I}_d - \mathbf{U}_k)\tilde{\boldsymbol{\epsilon}}$ is given by

$$\left(\mathbf{id} - \mathbf{U}_k\boldsymbol{f} - (\mathbf{I}_d - \mathbf{U}_k)\tilde{\boldsymbol{f}}\right)(\boldsymbol{X}). \qquad (6)$$

We now make the following assumption regarding the stability of the interventional experiments.

**Assumption 2.1** (Interventional solvability). For each intervention $I_k$ considered in this work, the forward map given by (6) is invertible.

Let $p_{\widetilde{E}}$ denote the probability density of $\tilde{\boldsymbol{\epsilon}}$. The probability distribution of $\boldsymbol{X}$ under the $k$-th experiment can be written as:

$$
\begin{aligned}
p_{I_k, \mathcal{G}}(\mathbf{X}) = p_E\Big( & \big[(\mathbf{id} - \mathbf{U}_k \boldsymbol{f})(\boldsymbol{X})\big]_{\mathcal{U}_k} \Big) \\
\times\, p_{\widetilde{E}}\Big( & \big[(\mathbf{id} - \tilde{\mathbf{U}}_k \tilde{\boldsymbol{f}})(\boldsymbol{X})\big]_{I_k} \Big) \\
\times\, & \big|\det(\mathbf{J}_{(\mathbf{id} - \boldsymbol{f}^{(I_k)})}(\boldsymbol{X}))\big|, \quad (7)
\end{aligned}
$$

where $\boldsymbol{f}^{(I_k)} \triangleq (\mathbf{U}_k \boldsymbol{f} + (\mathbf{I}_d - \mathbf{U}_k)\tilde{\boldsymbol{f}}), \mathcal{U}_k \triangleq V \setminus I_k$, is the combined intervened causal mechanism and $\tilde{\mathbf{U}}_k = \mathbf{I} - \mathbf{U}_k$.

We deal with three different types of soft interventions:

- **Shift Interventions**: The intervened noise variable is obtained by shifting the mean of the observed noise variable by a finite number, i.e., $\tilde{\epsilon}_i = \tilde{\mu}_i + \epsilon_i$.

- **Scale Interventions**: The intervened noise variable is obtained by scaling the variance of the observed noise variable by a finite number, i.e., $\tilde{\epsilon}_i = \tilde{\sigma}_i^2 \epsilon_i$.

- **Noisy Function Interventions**: The intervened causal mechanism is obtained by changing the structural parameters of the observed causal mechanism, provided that Assumption 2.1 is satisfied. .

Given data obtained from $K$ interventional experiments, *our goal in this work is to learn the structure of the cyclic causal graph as well as the interventional targets by maximizing the log-likelihood of the data.*

# 3. SCOUT: Cyclic Causal Discovery Under Soft Interventions with Unknown Targets

## 3.1. Using Normalizing Flows for Causal Learning under Non-Gaussian Noise

*Normalizing flows* are a class of generative models that are capable of transforming a simple distribution (standard Gaussian) to something more complex through a series of bijective transformations (Papamakarios et al., 2021). Within our framework, normalizing flows are employed twice: first we use *contractive residual flows* (Behrmann et al., 2019) to obtain the noise component $\boldsymbol{\epsilon}$ in the SEM of (2), then we use piecewise rational quadratic CDF transformation (Durkan et al., 2019) to transform this noise component $\boldsymbol{\epsilon}$ into a standard normal gaussian random vector $\boldsymbol{z}$ (allowing us to model more complex families of exogenous noise distributions).

### 3.1.1. MODELING THE CAUSAL FUNCTION

As mentioned in the Section 2, we assume that the forward mapping $\boldsymbol{X} \mapsto \boldsymbol{\epsilon} = (\mathbf{id} - \boldsymbol{f})(\boldsymbol{X})$ is invertible. According

to Banach's fixed point, we can satisfy this condition by restricting the function $\boldsymbol{f}$ to be *contractive* (see Appendix A for details).. A function $\boldsymbol{g} : \mathbb{R}^d \to \mathbb{R}^d$ is said to be contractive if there exists a constant $L < 1$ such that:

$$
\|\boldsymbol{g}(\boldsymbol{x}) - \boldsymbol{g}(\boldsymbol{y})\| \le L \|\boldsymbol{x} - \boldsymbol{y}\| \quad \text{for all } x, y \in \mathbb{R}^d.
$$

We use neural networks to parametrize the function $\boldsymbol{f}$, and the contractivity assumption can be conserved with spectral normalization of the network weights during each iteration. The adjacencies of the causal graph $\mathcal{G}$ can be introduced explicitly as a binary matrix $\boldsymbol{M}^{\mathcal{G}} \in \{0, 1\}^{d \times d}$, with 1 representing the presence an edge. As a result, the causal mechanism can be shown as:

$$
[\boldsymbol{f_\theta}(\boldsymbol{x})]_i = \big[\mathrm{NN}_{\boldsymbol{\theta}}\big(\boldsymbol{M}_{*,i}^{\mathcal{G}} \odot \boldsymbol{x}\big)\big]_i, \quad (8)
$$

where $\mathrm{NN}_{\boldsymbol{\theta}}$ denotes a fully connected neural network parameterized by $\boldsymbol{\theta}$, $\odot$ denotes the Hadamard product, and $\boldsymbol{M}_{*,i}^{\mathcal{G}}$ is the $i$-th column of $\boldsymbol{M}^{\mathcal{G}}$. The entries of $\boldsymbol{M}^{\mathcal{G}}$ are sampled from the Gumbel-softmax distribution $\boldsymbol{M}_\phi$ (Jang et al., 2017) and the parameters $\phi$ are updated during training using straight-through gradient estimation. $\lambda_{\mathcal{G}} \mathbb{E}_{\boldsymbol{M}^{\mathcal{G}} \sim \boldsymbol{M}_\phi}[\|M\|_1]$ will be used as the regularizer in the loss function to favor a sparse adjacency matrix.

In the case of soft interventions, we model $\tilde{\boldsymbol{f}}$ similar to $\boldsymbol{f}$ with another set of neural network parameters $\tilde{\boldsymbol{\theta}}$ while preserving the same adjacency matrix $\boldsymbol{M}^{\mathcal{G}}$ that is:

$$
[\tilde{\boldsymbol{f}}_{\tilde{\theta}}(\boldsymbol{x})]_i = \big[\mathrm{NN}_{\tilde{\boldsymbol{\theta}}}\big(\boldsymbol{M}_{*,i}^{\mathcal{G}} \odot \boldsymbol{x}\big)\big]_i. \quad (9)
$$

While training the model, we rescale the weights of neural network layers of $\boldsymbol{f}$ and $\tilde{\boldsymbol{f}}$ to ensure they remain contractive.

### 3.1.2. TRANSFORMING THE NON-GAUSSIAN NOISE

Under the assumption that the noise vector $\boldsymbol{\epsilon}$ may be non-Gaussian, we require a tractable and efficient way to compute the noise distribution $p_E$ from (3). This can again be achieved using normalizing flows and assuming an invertible forward map $\boldsymbol{\epsilon} \mapsto \boldsymbol{z} = \boldsymbol{g}(\boldsymbol{\epsilon})$. Under this assumption, the probability density function for $\boldsymbol{\epsilon}$ is well-defined and can be written as:

$$
p_E(\boldsymbol{\epsilon}) = p_Z(\boldsymbol{g}(\boldsymbol{\epsilon})) \big|\det(\mathbf{J}_{\boldsymbol{g}}(\boldsymbol{\epsilon}))\big|, \quad (10)
$$

where $p_Z(\boldsymbol{z}) = \mathcal{N}(\boldsymbol{z} \mid \boldsymbol{0}, \boldsymbol{I})$ denotes the probability density function of standard normal distribution and $\mathbf{J}_{\boldsymbol{g}}(\boldsymbol{\epsilon})$ denotes the Jacobian matrix of the function $\boldsymbol{g}$ evaluated at $\boldsymbol{\epsilon}$.

Since, in Section 2, we assumed our causal system is free of confounders, the noise samples will be independent. To preserve the independence assumption, we pick $z_i = g_i(\epsilon_i)$. This also simplifies the Jacobian $\mathbf{J}_{\boldsymbol{g}}(\boldsymbol{\epsilon})$ to a simple diagonal matrix. We restrict $\mathbf{g}$ to be a piecewise rational quadratic CDF because of its expressivity in a wide family of distributions while being invertible and yielding a tractable Jacobian

determinant (Durkan et al., 2019). The map $g : \epsilon \mapsto z$ is modeled using neural networks.

For the noise vector under interventions, $\tilde{\epsilon}$, we define another forward function $\tilde{g}$ mapping it to standard Gaussian random variable. The overall transformation under intervention $I_k$ is given by: $z^{(I_k)} = \mathbf{U}_k g(\epsilon) + (\mathbf{I} - \mathbf{U}_k)\tilde{g}(\tilde{\epsilon})$.

### 3.2. Finding Unknown Intervention Targets

In order to identify the unknown interventional targets for each setting $I_k \in \mathcal{I}$, we define an interventional target matrix $\boldsymbol{T}^{\mathcal{I}} \in \{0, 1\}^{K \times d}$ where each row denotes the specific experiment and each column depicts the nodes in the graph. $(\boldsymbol{T})_{kj}^{\mathcal{I}} = 0$ indicates that $X_j$ is intervened on in the $k$-th interventional experiment. The entries of $\boldsymbol{T}^{\mathcal{I}}$ are sampled using Gumbel-softmax distribution $\boldsymbol{T}_{\psi}$. Similar to graph adjacency learning, the parameters $\psi$ are updated during training using straight-through gradient estimation. We also introduce another regularizer favoring sparse intervention targets calculated as $\lambda_{\mathcal{I}} \mathbb{E}_{\boldsymbol{T}^{\mathcal{I}} \sim \boldsymbol{T}_{\psi}} \left[ \|\boldsymbol{T}^{\mathcal{I}}\|_1 \right]$.

### 3.3. Computing the log-determinant of the Jacobian

The computation of the log-determinant of the Jacobian term $\log\left|\det(\mathbf{J}_{(\mathbf{id}-f^{(I_k)})}(\boldsymbol{X}))\right|$ is a significant challenge. To address this issue, we use the unbiased estimator introduced by Behrmann et al. (2019), which is based on the following power series expansion:

$$
\log\left|\det \mathbf{J}_{(\mathbf{id}-f^{(I_k)})}(\boldsymbol{X})\right| = \log\left|\det\left(\mathbf{I} - \mathbf{J}_{(f^{(I_k)})}(\boldsymbol{X})\right)\right|
$$
$$
= -\sum_{m=1}^{\infty} \frac{1}{m} \operatorname{Tr}\left\{ \mathbf{J}_{(f^{(I_k)})}^m(\boldsymbol{X}) \right\},
$$
(11)

where $\mathbf{I} \in \mathbb{R}^{d \times d}$ denotes the identity matrix. The contractivity of $f^{(I_k)}$ guarantees the convergence of the above series. The trace term can be further simplified using the Hutchinson trace estimator:

$$
\operatorname{Tr}\left\{ \mathbf{J}_{(f^{(I_k)})}^m(\boldsymbol{X}) \right\} = \mathbb{E}_{\mathbf{w}}\left[ \mathbf{w}^\top \mathbf{J}_{(f^{(I_k)})}^m(\boldsymbol{X}) \, \mathbf{w} \right],
$$
(12)

where $\mathbf{w} \sim \mathcal{N}(\mathbf{0}, \mathbf{I})$. In practice, the above power series can be evaluated by truncating it to a finite number of terms. However, this truncation causes the estimator to be biased; to make it unbiased, we follow the method proposed by Chen et al. (2019). We truncate the series to a random cut-off $n \sim p(N)$, where $p$ is a probability distribution over natural numbers $\mathbb{N}$. In this work, we pick $p$ to be the Poisson distribution $n \sim \operatorname{Poi}(N)$, where we treat $N$ as a hyperparameter. Finally, each term in the series is reweighted to obtain the following estimator for the log-determinant of the Jacobian:

$$
\log\left|\det \mathbf{J}_{(\mathbf{id}-f^{(I_k)})}(\boldsymbol{X})\right| = -\mathbb{E}_{n,\mathbf{w}}\left[ \sum_{m=1}^{n} \frac{\mathbf{w}^\top \mathbf{J}_{(f^{(I_k)})}^m \mathbf{w}}{m \cdot \mathbb{P}(N \geq k)} \right].
$$
(13)

### 3.4. The Score Function

Our primary objective in this work is to determine the parameters of the SEM, specifically the causal graph structure, causal mechanism, and intervention targets for each experiment. To that end, as in previous works, we use regularized log-likelihoods as the score function to be maximized. Given a candidate graph $\mathcal{G}$ and a set of interventional targets $\mathcal{I}$, the score function can be written as:

$$
\mathcal{S}(\mathcal{G}, \mathcal{I}) = \sup_{\boldsymbol{\theta}} \sum_{k=1}^{K} \mathbb{E}_{\boldsymbol{X} \sim p^{(k)}}\left[ \log p_{I_k, \mathcal{G}}(\boldsymbol{X}) \right]
$$
$$
- \lambda_{\mathcal{G}}|\mathcal{G}| - \lambda_{\mathcal{I}}|\mathcal{I}|,
$$
(14)

where $p^{(k)}$ is the data generating distribution for the $k$-th experiment, $\log p_{I_k, \mathcal{G}}(\boldsymbol{X})$ is given by (7), $\lambda_{\mathcal{G}}$ and $\lambda_{\mathcal{I}}$ terms are the regularizers discussed in sections 3.1.1 and 3.2 respectively, and $\boldsymbol{\theta}$ is the causal system parameters.

We will now present the main theoretical result of this paper. This theorem will establish that under certain assumptions, exact maximizing the score function given in (14) with respect to $\mathcal{G}$ and $\mathcal{I}$ will recover the $\mathcal{I}^*$-Markov equivalence class of $\mathcal{G}^*$ and the ground truth interventional family $\mathcal{I}^*$. Due to space constraints, a detailed proof of this theorem will be provided in the Appendix B.

**Theorem 3.1.** *Let $\mathcal{G}^*$ be the ground truth graph and let $\mathcal{I}^*$ be the ground truth intervention family. $(\hat{\mathcal{G}}, \hat{\mathcal{I}}) \in \arg\max_{\mathcal{G}, \mathcal{I}} \mathcal{S}(\mathcal{G}, \mathcal{I})$. Under the Assumptions 2.1, B.8, B.9, B.10 and B.11, and for suitable $\lambda_{\mathcal{G}}, \lambda_{\mathcal{I}} > 0$, $\hat{\mathcal{G}}$ is $\mathcal{I}^*$-Markov equivalent to $\mathcal{G}^*$ and $\hat{\mathcal{I}} = \mathcal{I}^*$.*

*Proof (Sketch).* Using the characterization of general directed Markov equivalence class from Bongers et al. (2021), augmenting it to include interventions, we show that any graph $\mathcal{G}$ outside of this equivalence class or any intervention family $\mathcal{I} \neq \mathcal{I}^*$ will yield a strictly lower score than the $\mathcal{S}(\mathcal{G}^*, \mathcal{I}^*)$. This can be demonstrated by showing that the augmented graph built from a graph outside this equivalence class and an incorrect intervention family either misses certain existing independencies or imposes extra independencies that do not exist in the data. Furthermore, coefficients $\lambda_{\mathcal{G}}, \lambda_{\mathcal{I}}$ should be chosen small enough to avoid too much sparse solutions. $\square$

The score function in (14) is defined under the infinite data limit. To make it computable under finite data samples, we

redefine it in the following way:

$$\hat{\mathcal{S}}(\boldsymbol{M}_\phi, \boldsymbol{T}_\psi) = \sup_\theta \mathbb{E}_{\substack{\mathbf{M}^{\mathcal{G}} \sim \boldsymbol{M}_\phi \\ \mathbf{T}^{\mathcal{I}} \sim \boldsymbol{T}_\psi}} \left[ \sum_{k=1}^{K} \sum_{i=1}^{N_k} \log p_{I_k, \mathcal{G}}(\boldsymbol{x}^{(i,k)}) \right.$$
$$\left. - \lambda_{\mathcal{G}} \|\boldsymbol{M}\|_1 - \lambda_{\mathcal{I}} \|\boldsymbol{T}\|_1 \right]. \tag{15}$$

where we take a summation over finite samples $\boldsymbol{x}^{(i,k)}$ of each experiment instead of taking the expectation over the data distribution. We optimize the score function (15) with respect to the neural network parameters $\theta$, the graph structure parameters $\boldsymbol{M}_\phi$ and intervention target parameters $\boldsymbol{T}_\psi$.

## 4. Experiments

The code for SCOUT is available at the repository: https://github.com/alparturkoglu/scout-master

We evaluated SCOUT on both synthetic and real-world datasets. We also compared its performance against existing state-of-the-art causal discovery algorithms, NODAGS-Flow (Sethuraman et al., 2023), LLC (Hyttinen et al., 2012), and BACKSHIFT (Rothenhäusler et al., 2015). NODAGS-Flow can learn nonlinear cyclic causal graphs with interventions; however, it assumes that the interventions are surgical with the interventional targets being known. Additionally, NODAGS-Flow assumes the exogenous noise to be Gaussian. LLC does not assume a normal noise distribution; however, it is limited to Linear SEM and requires interventional targets to operate. BACKSHIFT operates under unknown targets and deals with shift interventions; however, it cannot handle other types of soft interventions. Furthermore, it is also designed to work with Linear SEM. We also provide a comparison between SCOUT and other baselines which can handle nonlinearity, soft-interventions, and unknown targets, but specifically for DAGs in Appendix C.1,

### 4.1. Synthetic data

We generated cyclic graphs using the Erdős-Rényi (ER) random graph model with $d = 10$ nodes and outgoing edge density of 2. Our training data consists of 10 experiments, one for each single-node intervention, and each experiment contains $1,000$ samples. For all the experimental results presented here, we used non-linear SEMs constrained to be contractive. We used nine different settings in our experiments, varying the exogenous noise variable between Gaussian, Exponential, and Gumbel distributions, and varying the soft intervention type between shift, scale, and noisy function. When a node is not intervened, the corresponding exogenous noise distribution parameters are set as follows: for the Gaussian noise setting, the noise mean is set to 0, and variance is set to 0.25. For the exponential noise setting

the rate is set to 2. Finally, for the Gumbel noise setting the location and scale of the distribution is set to 0 and 0.5 respectively. We set the shift and scale parameters to 2 for the respective interventions, and for the noisy-function interventions, we negate the causal mechanism.

The performance on synthetic data is evaluated with respect to both graph structure recovery and unknown intervention target recovery. We compare the learned adjacency matrix with the binary ground truth adjacency matrix to evaluate the graph recovery, and we compare the learned intervention targets to the binary ground truth interventional target matrix. We use the Area Under Precision-Recall Curve (AUPRC) as the error metric. AUPRC computes the area under the precision-recall curve evaluated at various threshold values (the higher the better). The results of the synthetic experiments are presented in Figure 1 and Table 1.

The box plot in Figure 1 shows the median and interquartile range of the AUPRC metric for all the models over ten independent trials. Each plot in Figure 1 shows the performance of our framework compared to baselines with respect to the interventional setting given at the left of the row, i.e., the vertical labels, and the exogenous noise presented at the top of the column. As seen from Figure 1, SCOUT acheives near perfect graph recovery in all settings except Noisy Function + Gaussian noise (where it attains comparable performance to that of the baseline methods).

Table 1 presents the intervention target recovery performance of SCOUT compared to BACKSHIFT, providing mean AUPRC values along with standard deviations in the same settings as Figure 1. The other baselines do not identify unknown targets; therefore, they are not included in this table. For scale interventions, both models can identify targets perfectly, matching the results in the Figure 1. For shift interventions, SCOUT can recover the targets perfectly, whereas BACKSHIFT achieves comparable performance; however, it is insufficient to recover the ground truth graph, as shown again in Figure 1. While both SCOUT and BACKSHIFT fail to fully recover the interventional targets for noisy function interventions, SCOUT still outperforms BACKSHIFT in this setting. This is expected since the noisy function interventions can not induce a significant distribution change compared to shift/scale interventions (see the Appendix C.6 for numerical results), and in practice, identifying intervention targets typically requires interventions that induce sufficiently strong distributional changes for finite data (Gamella & Heinze-Deml, 2020).

#### 4.1.1. SCALING WITH NODES

We compare the performance of SCOUT to the baselines as the number of nodes varies from 10 to 70. We look at nonlinear SEM under unknown shift and scale interventions with Gaussian and Gumbel noise. As shown in Figure 2 and

*Table 1.* Interventional target recovery performance comparison between SCOUT and BACKSHIFT under non-linear SEM and various interventional and exogenous noise settings, evaluated using AUPRC (higher is better). Results are reported as mean $\pm$ standard deviation over 10 independent trials. In all cases, the number of nodes is fixed at $d = 10$.

| | Gaussian | | Exponential | | Gumbel | |
|---|---|---|---|---|---|---|
| Intervention Type | SCOUT | BACKSHIFT | SCOUT | BACKSHIFT | SCOUT | BACKSHIFT |
| Shift | $1.000 \pm 0.000$ | $0.962 \pm 0.038$ | $1.000 \pm 0.000$ | $0.868 \pm 0.116$ | $1.000 \pm 0.000$ | $0.859 \pm 0.144$ |
| Scale | $1.000 \pm 0.000$ | $1.000 \pm 0.000$ | $0.975 \pm 0.079$ | $1.000 \pm 0.000$ | $1.000 \pm 0.000$ | $1.000 \pm 0.000$ |
| Noisy Function | $0.264 \pm 0.133$ | $0.259 \pm 0.107$ | $0.590 \pm 0.123$ | $0.313 \pm 0.214$ | $0.607 \pm 0.152$ | $0.423 \pm 0.269$ |

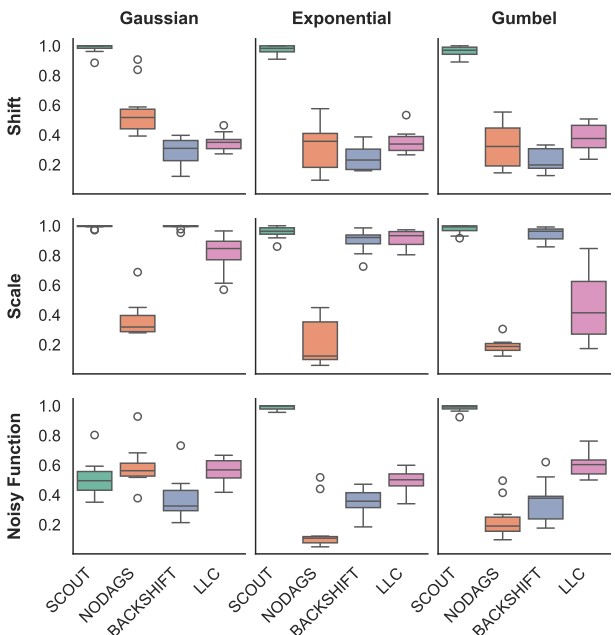

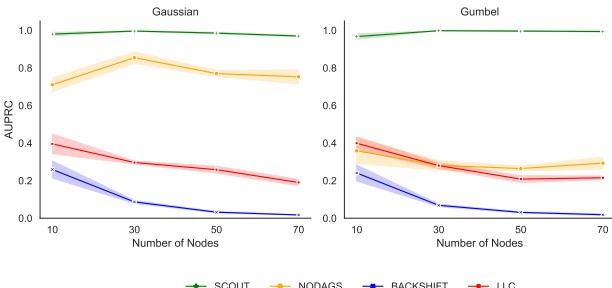

*Figure 2.* Graph recovery performance comparison between SCOUT and baselines under non-linear SEM and shift interventions. The number of nodes is varied from $d = 10$ to $70$

*Table 2.* Interventional target recovery performance comparison between SCOUT and BACKSHIFT under non-linear SEM and shift interventions. The table presents the average AUPRC value along with its standard deviation evaluated using 5 independent trials. The number of nodes is varied from $d = 10$ to $70$.

| | Gaussian | | Gumbel | |
|---|---|---|---|---|
| $d$ | SCOUT | BACKSHIFT | SCOUT | BACKSHIFT |
| 10 | $1.00\pm0.00$ | $0.90\pm0.14$ | $1.00\pm0.00$ | $0.87\pm0.13$ |
| 30 | $1.00\pm0.00$ | $0.51\pm0.15$ | $1.00\pm0.00$ | $0.51\pm0.14$ |
| 50 | $1.00\pm0.00$ | $0.08\pm0.02$ | $1.00\pm0.00$ | $0.05\pm0.01$ |
| 70 | $1.00\pm0.00$ | $0.03\pm0.00$ | $1.00\pm0.00$ | $0.02\pm0.00$ |

*Figure 1.* Graph recovery performance comparison between SCOUT and baselines under non-linear SEM and various interventional and exogenous noise settings, evaluated using AUPRC (the higher the better). The box plots show the median and interquartile ranges across ten independent trials. In all cases, the number of nodes is fixed at $d = 10$.

3, SCOUT's structure recovery performance remains relatively high, whereas other baselines yield lower AUPRCs as the graph size increases. Table 2 suggests that while BACKSHIFT begins not to detect intervention targets correctly, SCOUT still achieves a perfect performance in terms of interventional target recovery for large-scale graphs. Overall, these results show that SCOUT is highly scalable as graph sizes increase.

### 4.1.2. EFFECT OF NEURAL SPLINE FLOWS

To assess the contribution of each component in our model, we perform an ablation study comparing SCOUT, SCOUT-noNSF, and NODAGS (Sethuraman et al., 2023). In SCOUT-noNSF, the Neural Spline Flow (NSF) layer is replaced with a simple Gaussian likelihood. NODAGS is designed for Gaussian noise and hard interventions with known targets.

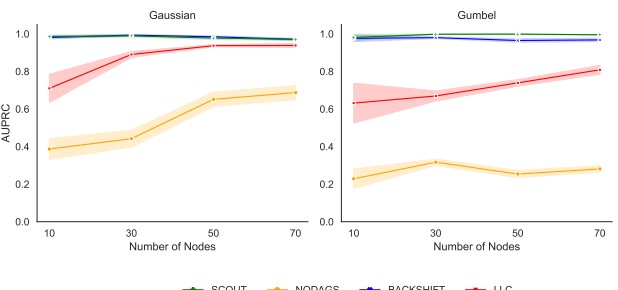

*Figure 3.* Graph recovery performance comparison between SCOUT and baselines under non-linear SEM and scale interventions. The number of nodes is varied from $d = 10$ to $70$

While SCOUT-noNSF retains the ability to model soft interventions and includes an interventional target matrix for handling unknown targets, it lacks the flexibility to transform non-Gaussian noise distributions. In contrast, SCOUT incorporates the NSF layer, enabling it to map arbitrary

noise distributions to a standard Gaussian space, thereby improving identifiability and learning.

We conduct this experiment under the same setting as Figure 1. As shown in Figure 4 and Table 3, removing the NSF layer leads to a significant degradation in performance: intervention targets become unidentifiable, and graph recovery fails in the unknown-target setting.

However, as shown in Figure 5, when the intervention targets are known and the noise is Gaussian, the contractive residual structure alone is sufficient for accurate graph recovery.

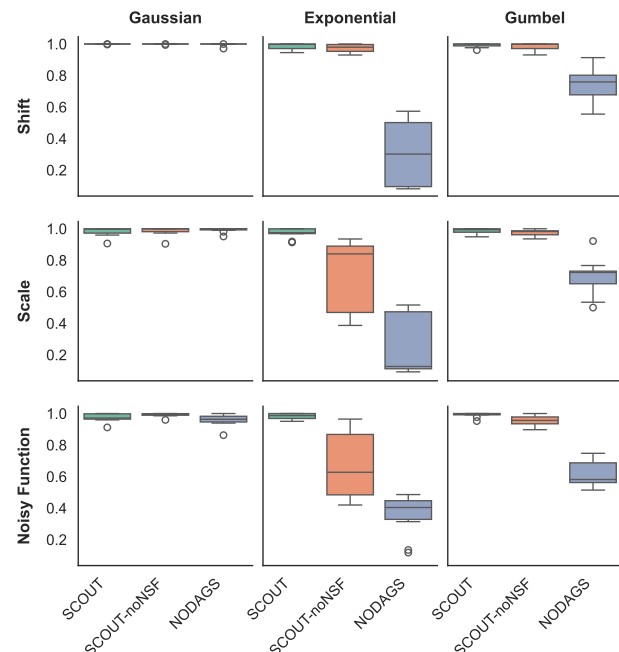

*Figure 5.* Graph recovery performance comparison between SCOUT, SCOUT-noNSF, and NODAGS for known intervention targets under nonlinear SEM and various interventional and exogenous noise settings, evaluated using AUPRC. In all cases, the number of nodes is fixed at $d = 10$.

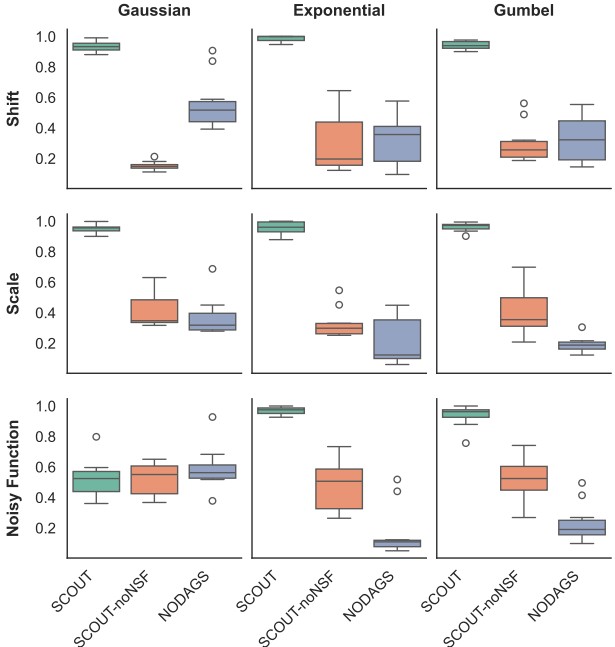

*Figure 4.* Graph recovery performance comparison between SCOUT, SCOUT-noNSF, and NODAGS under nonlinear SEM and various interventional and exogenous noise settings, evaluated using AUPRC. In all cases, the number of nodes is fixed at $d = 10$.

Additional experiments, including performance evaluations on non-contractive SEMs (DAGs) in Appendix C.1, linear SEMs in Appendix C.2, hard interventions in Appendix C.4, and known intervention targets in Appendix C.3, are provided in Appendix C. Further ablation studies assessing the robustness of the model are presented in Appendix C.5.

## 4.2. Real World Data

### 4.2.1. GENE REGULATORY NETWORKS (PERTURB-CITE-SEQ)

We evaluate SCOUT's performance on learning the causal graph structure of gene regulatory networks from real-world gene expression data with genetic interventions. We focus on the PerturbCITE-seq dataset (Frangieh et al., 2021) that

contains gene expressions taken from 218,331 melanoma cells split over three different cell conditions: (i) control, (ii) co-culture, and (iii) IFN-$\gamma$.

Due to computational limitations, we limit our analysis to 61 genes out of approximately 20,000 in the genome, and we treat each cell condition as a separate dataset, following the setup of (Sethuraman et al., 2023). We train SCOUT, along with the baselines, on these three datasets by supplying single-gene interventions for the 61 genes as unknown targets. The adjacency matrix recovered by SCOUT for the cell condition co-culture is given in Figure 23.

Since the dataset does not include a ground-truth causal graph, we evaluate the performance of SCOUT and baselines based on predictive performance under unseen interventions. We perform a 90-10 split of the dataset, taking 90% of the data as training set and the remaining 10% as the testing set. As a performance metric, we use negative log-likelihood (NLL) on the test portion of the data after training the model (lower the better). Since BACKSHIFT does not have a method to compute this metric, it is not included in the results. The results can be seen in Figure 6. SCOUT outperforms all baselines across all cell conditions, demonstrating that accounting for non-Gaussian exogenous noise enables the model to learn the target distribution better and improve its predictive power. Additionally, we present a performance comparison of SCOUT with other baselines

*Table 3.* Interventional target recovery performance comparison between SCOUT and SCOUT without NSF under non-linear SEM and various interventional and exogenous noise settings, evaluated using AUPRC. Results are reported as mean $\pm$ standard deviation over 10 independent trials. In all cases, the number of nodes is fixed at $d = 10$.

| Intervention Type | Gaussian | | Exponential | | Gumbel | |
|---|---|---|---|---|---|---|
| | SCOUT | SCOUT-noNSF | SCOUT | SCOUT-noNSF | SCOUT | SCOUT-noNSF |
| Shift | $1.000 \pm 0.000$ | $0.144 \pm 0.049$ | $1.000 \pm 0.000$ | $0.471 \pm 0.145$ | $1.000 \pm 0.000$ | $0.237 \pm 0.078$ |
| Scale | $1.000 \pm 0.000$ | $0.227 \pm 0.077$ | $0.975 \pm 0.079$ | $0.232 \pm 0.128$ | $1.000 \pm 0.000$ | $0.171 \pm 0.070$ |
| Noise | $0.264 \pm 0.133$ | $0.393 \pm 0.183$ | $0.590 \pm 0.123$ | $0.443 \pm 0.211$ | $0.607 \pm 0.152$ | $0.463 \pm 0.244$ |

under known targets, as well as another comparison with respect to the mean absolute error (MAE) metric (including BACKSHIFT) in the Appendix C.7.

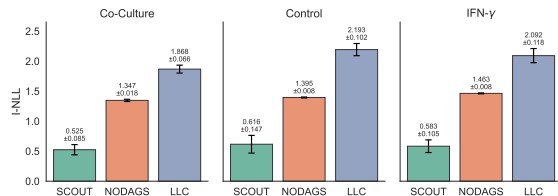

*Figure 6.* The performance comparison results on the Perturb-CITE-seq (Frangieh et al., 2021) gene perturbation dataset. The bar graph shows interventional negative log-likelihood (I-NLL) and its standard deviation across 5 trials.

### 4.2.2. CLOSE-TO-REAL-WORLD DATA (SERGIO)

Although experiments on the real-world Perturb-CITE-seq dataset show that SCOUT outperforms the baselines in terms of NLL, this dataset does not provide a ground-truth causal graph. To evaluate the graph recovery and unknown intervention target recovery performance of SCOUT in a close-to-real-world setting, we use SERGIO (Dibaeinia & Sinha, 2020), a simulator for single-cell gene expression data generated from a gene regulatory network (GRN). SERGIO produces realistic stochastic expression data using non-linear regulatory dynamics, including Hill-type effects, and can simulate perturbations such as gene knockouts across different environments and cell types.

We set the number of genes (nodes) to $d = 10$. The underlying random cyclic graph is sampled with edge probability $p = 0.25$ and contains 2 master regulators, i.e., nodes with no incoming edges. The SERGIO simulation hyperparameters are set to `noise_params`=1.0, `decay`=0.8, `hill`=2, and `sampling_state`=4. For each environment, we simulate 1000 cells, with $N_{\text{types}} = 5$ cell types and $N_{\text{cells/type}} = 200$ cells per type. We again use a single-node intervention design, namely one single-gene knockout environment per node, resulting in $K = 10$ intervention environments.

The results in Figure 7 show that SCOUT outperforms the existing baselines in graph identification, although it

does not achieve perfect recovery because the overall causal mechanism generated by SERGIO need not be contractive. In addition, SCOUT achieves near-perfect intervention target recovery, with an AUPRC of $0.978 \pm 0.025$, compared to $0.432 \pm 0.032$ for BACKSHIFT.

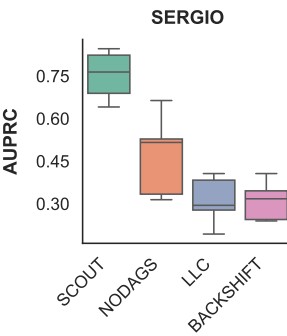

*Figure 7.* The performance comparison results on SERGIO evaluated using AUPRC.

## 5. Discussion and Conclusion

We proposed SCOUT, a novel framework for causal discovery that simultaneously infers directed cyclic causal structure and unknown intervention targets from soft interventional data under non-Gaussian noise. It models causal relationships as neural networks and recovers the ground-truth graph along with intervention targets via likelihood score maximization. We provided consistency proof for the recovery of the Markov equivalence class of the ground truth graph under unknown intervention targets. We conducted experiments on both synthetic and real-world data to demonstrate that SCOUT outperforms state-of-the-art methods in causal graph recovery and identification of target nodes that are intervened on, across various interventional and noise settings. We showed that our model is highly scalable with increasing graph size and maintains its robustness with increasing number of intervention targets. Evaluations on the Perturb-CITE-seq dataset show that our model also achieves superior predictive accuracy in real-world scenarios. Possible research directions to extend this work include incorporating more realistic measurement noise models, allowing the system to handle confounders, or scaling up to handle larger graph models.

## Acknowledgments

This material is based on work supported National Science Foundation (NSF) under grant number 2502298.

## Impact Statement

This paper presents work whose goal is to advance the field of Machine Learning. There are many potential societal consequences of our work, none which we feel must be specifically highlighted here.

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

v108/zheng20a.html.

The appendix is organized as follows: Appendix A discusses the solvability of cyclic systems under equilibrium and justifies the contractiviy assumption on observed and intervened causal mechanisms. Appendix B develops the theoretical basis for the cyclic causal discovery under soft interventions with unknown targets, including the proof of Theorem 3.1 and a characterization of the score-maximizing equivalence class of directed graphs. Appendix C reports additional experimental results comparing SCOUT against these baselines. Appendix D details the experimental setup and implementation of SCOUT and the baseline methods.

## A. Solvability of Cyclic Systems Under Equilibrium

Our causal discovery framework relies on the existence of unique observational and interventional distributions, as well as Markov properties with respect to $\sigma$-/$d$-separation. These requirements are automatically satisfied for acyclic SCMs. However, when cycles are allowed, stronger conditions are needed to guarantee unique solvability. In particular, (Bongers et al., 2021) showed that unique solvability with respect to each strongly connected component is necessary for an SCM to satisfy the Markov property with respect to $\sigma$-separation. The same work also characterizes a class of SCMs, called *simple SCMs*, that satisfy these unique solvability requirements. This assumption is also standard in several constraint-based causal discovery methods (M. Mooij & Claassen, 2020; Mokhtarian et al., 2023).

This motivates our restriction to contractive causal mechanisms. Contractivity guarantees unique solvability and the Markov property with respect to $\sigma$-separation. Therefore, the SCMs considered in our work form a subset of simple SCMs.

Under soft interventions, however, an SCM need not remain simple. In other words, the intervened system may fail to admit a unique interventional distribution. In that case, the Markov property is no longer guaranteed, and causal discovery from static equilibrium data is no longer a well-posed problem.

We assume that, in the observational regime, the dynamical system evolves according to

$$\mathbf{x}^{(t)} := f\Big(\mathbf{x}^{(t-1)}\Big) + \boldsymbol{\varepsilon}.$$

If $f$ is contractive, then by the Banach fixed-point theorem, for every initial value $\mathbf{x}^{(0)}$ and every $\boldsymbol{\varepsilon}$, the sequence $\{\mathbf{x}^{(t)}\}_{t \geq 0}$ converges to a unique fixed point $\mathbf{x}^\star$ satisfying

$$\mathbf{x}^\star = f(\mathbf{x}^\star) + \boldsymbol{\varepsilon}.$$

Hence, the observed equilibrium $\mathbf{X}$ satisfies

$$\boldsymbol{\varepsilon} = (\mathrm{id} - f)(\mathbf{X}).$$

Similarly, under an intervention that changes either the mechanism or the noise characteristics, the dynamical system can be written as

$$\mathbf{x}^{(t)} = \mathbf{U}_k\big(f(\mathbf{x}^{(t-1)}) + \boldsymbol{\varepsilon}\big) + (\mathbf{I} - \mathbf{U}_k)\big(\tilde{f}(\mathbf{x}^{(t-1)}) + \tilde{\boldsymbol{\varepsilon}}\big),$$

where $\mathbf{U}_k$ is a diagonal matrix with ones corresponding to non-intervened variables and zeros corresponding to intervened variables, and $\mathbf{I}$ denotes the identity matrix. Equivalently, defining the intervened mechanism and effective noise as

$$f^{(I_k)}(\mathbf{x}^{(t-1)}) := \mathbf{U}_k f(\mathbf{x}^{(t-1)}) + (\mathbf{I} - \mathbf{U}_k)\tilde{f}(\mathbf{x}^{(t-1)}), \qquad \boldsymbol{\eta} := \mathbf{U}_k \boldsymbol{\varepsilon} + (\mathbf{I} - \mathbf{U}_k)\tilde{\boldsymbol{\varepsilon}},$$

we can write

$$\mathbf{x}^{(t)} := f^{(I_k)}\Big(\mathbf{x}^{(t-1)}\Big) + \boldsymbol{\eta}.$$

If $f^{(I_k)}$ is contractive, then again by the Banach fixed-point theorem, for every initialization, the iterates converge to a unique fixed point $\mathbf{x}_k^\star$ satisfying

$$\mathbf{x}_k^\star = f^{(I_k)}(\mathbf{x}_k^\star) + \boldsymbol{\eta}.$$

Therefore, the observed interventional equilibrium $\mathbf{X}$ satisfies

$$\boldsymbol{\eta} = \big(\mathrm{id} - f^{(I_k)}\big)(\mathbf{X}).$$

Thus, if the intervened mechanism $f^{(I_k)}$ remains contractive, the Banach fixed-point theorem guarantees that the intervention admits a unique equilibrium.

# B. Theory

In this section, we lay out the theory behind cyclic causal discovery under soft interventions with unknown targets. We start by summarizing the definitions and establish results required for the proof of Theorem 3.1, beginning with standard graph-theoretic notation.

## B.1. Preliminaries

Consider a directed graph $\mathcal{G} = (\mathcal{V}, \mathcal{E})$. A *path* $\pi$ between nodes $i$ and $j$ is a sequence $(i_0, \varepsilon_1, i_1, \ldots, \varepsilon_n, i_n)$, where $\{i_0, \ldots, i_n\} \subseteq \mathcal{V}$ and $\{\varepsilon_1, \ldots, \varepsilon_n\} \subseteq \mathcal{E}$, with $i_0 = i$ and $i_n = j$. A path is *directed* if each edge $\varepsilon_k$ follows the form $i_{k-1} \to i_k$ for all $k \in [n]$. A cycle through node $i$ consists of a directed path from $i$ to some node $j$ and an additional edge $j \to i$. For any node $i \in \mathcal{V}$, the *ancestor set* is defined as $\mathrm{an}_{\mathcal{G}}(i) := \{j \in \mathcal{V} \mid$ a directed path from $j$ to $i$ exists in $\mathcal{G}\}$, while the *descendant set* is given by $\mathrm{de}_{\mathcal{G}}(i) := \{j \in \mathcal{V} \mid$ a directed path from $i$ to $j$ exists in $\mathcal{G}\}$. The *strongly connected component* of $i$, denoted $\mathrm{sc}_{\mathcal{G}}(i)$, is the intersection of its ancestors and descendants: $\mathrm{sc}_{\mathcal{G}}(i) = \mathrm{an}_{\mathcal{G}}(i) \cap \mathrm{de}_{\mathcal{G}}(i)$.

**Definition B.1** (Collider). For a directed graph $\mathcal{G} = (\mathcal{V}, \mathcal{E})$, a node $i_k \in \mathcal{V}$ in a path $\pi = (i_0, \varepsilon_1, i_1, \varepsilon_2, \ldots, i_{n-1}, \varepsilon_n, i_n)$ is called a *collider* if $k \neq 0, n$ (non-endpoint) and the two edges $\varepsilon_k, \varepsilon_{k+1}$ have their heads pointed at $i$, i.e., the subpath $(i_{k-1}, \varepsilon_k, i_k, \varepsilon_{k+1}, i_{k+1})$ is of the form $i_{k-1} \to i_k \leftarrow i_{k+1}$. The node $i_k$ is called a *non-collider* if $i_k$ is not a collider.

**Definition B.2** (d-separation). Let $\mathcal{G} = (\mathcal{V}, \mathcal{E})$ be a directed graph and let $C \subseteq \mathcal{V}$ be a subset of nodes. A path $\pi = (i_0, \varepsilon_1, i_1, \varepsilon_2, \ldots, i_{n-1}, \varepsilon_n, i_n)$ is said to be *d-blocked* given $C$ if

1. $\pi$ contains a collider $i_k \notin \mathrm{an}_{\mathcal{G}}(C)$

2. $\pi$ contains a non-collider $i_k \in C$.

The path $\pi$ is said to be *d-open* given $C$ if it is not *d*-blocked. Two subsets of nodes $A, B \subseteq \mathcal{V}$ is said to be *d*-separated given $C$ if all paths between $a$ and $b$, where $a \in A$ and $b \in B$, is *d*-blocked given $C$, and is denoted by

$$A \stackrel{d}{\underset{\mathcal{G}}{\perp}} B \mid C.$$

If the underlying graph is acyclic, *d*-separation implies conditional independence. That is, for subsets of nodes $A, B, C \subseteq \mathcal{V}$,

$$A \stackrel{d}{\underset{\mathcal{G}}{\perp}} B \mid C \implies \boldsymbol{X}_A \underset{p_{\mathcal{G}}}{\perp} \boldsymbol{X}_B \mid \boldsymbol{X}_C,$$

where $\perp_{p_{\mathcal{G}}}$ denotes conditional independence, and $p_{\mathcal{G}}$ denotes the observational distribution. This is known as the *directed global Markov property* of $\mathcal{G}$ (Forré & Mooij, 2017). However, in general, cyclic graphs do not obey the directed global Markov property as shown by (Bongers et al., 2021; Spirtes, 2013).

(Forré & Mooij, 2017) proposed $\sigma$-separation, a generalization of $d$-separation that extends the directed global Markov property to graphs with cycles.

**Definition B.3** ($\sigma$-separation). Let $\mathcal{G} = (\mathcal{V}, \mathcal{E})$ be a directed graph and let $C \subseteq \mathcal{V}$ be a subset of nodes. A path $\pi = (i_0, \varepsilon_1, i_1, \varepsilon_2, \ldots, i_{n-1}, \varepsilon_n, i_n)$ is said to be *$\sigma$-blocked* given $C$ if

1. the first node of $\pi$, $i_0 \in C$ or its last node $i_n \in C$, or

2. $\pi$ contains a collider $i_k \notin \mathrm{an}_{\mathcal{G}}(C)$

3. $\pi$ contains a non-collider $i_k \in C$ that points towards a neighbor that is not in the same strongly connected component as $i_k$ in $\mathcal{G}$, i.e, such that $i_{k-1} \leftarrow i_k$ in $\pi$ and $i_{k-1} \notin \mathrm{sc}_{\mathcal{G}}(i_k)$, or $i_k \to i_{k+1}$ in $\pi$ and $i_{k+1} \notin \mathrm{sc}_{\mathcal{G}}(i_k)$.

The path $\pi$ is said to be *$\sigma$-open* given $C$ if it is not $\sigma$-blocked. Two subsets of nodes $A, B \subseteq \mathcal{V}$ is said to be $\sigma$-separated given $C$ if all paths between $a$ and $b$, where $a \in A$ and $b \in B$, is $\sigma$-blocked given $C$, and is denoted by

$$A \stackrel{\sigma}{\underset{\mathcal{G}}{\perp}} B \mid C.$$

Note that $\sigma$-separation reduces to $d$-separation for acyclic graphs, that is, when $\mathrm{sc}_{\mathcal{G}}(i) = \{i\}$ for all $i \in \mathcal{V}$.

With $\sigma$-separation in place, we can now state the generalized directed global Markov property.

**Definition B.4** (General directed global Markov property (Forré & Mooij, 2017))**.** Let $\mathcal{G} = (\mathcal{V}, \mathcal{E})$ be a directed graph and $p_{\mathcal{G}}$ denote the probability density of the observations $\boldsymbol{X}$. The probability density $p_{\mathcal{G}}$ satisfies the *general directed global Markov property* if for $A, B, C \subseteq \mathcal{V}$

$$A \overset{\sigma}{\underset{\mathcal{G}}{\perp}} B \mid C \implies \boldsymbol{X}_A \underset{p_{\mathcal{G}}}{\perp} \boldsymbol{X}_B \mid \boldsymbol{X}_C,$$

that is, $\boldsymbol{X}_A$ and $\boldsymbol{X}_B$ are conditionally independent given $\boldsymbol{X}_C$.

### B.2. Joint Causal Modelling and Markov properties

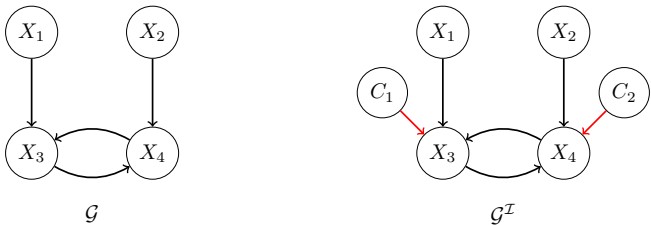

*Figure 8.* Illustration of the augmented graph $\mathcal{G}^{\mathcal{I}}$ corresponding to the set of interventional targets $\mathcal{I} = \{\emptyset, \{X_3\}, \{X_4\}\}$.

To integrate multiple interventional settings in a single causal graph, we adopt the idea of *joint causal model* proposed by (Mooij et al., 2016) by introducing a new set of context variables $\boldsymbol{C}^{\mathcal{I}} = (\boldsymbol{C}_1, \ldots, \boldsymbol{C}_K)$ each representing another interventional setting. (The scenario where $\boldsymbol{C}_k = \emptyset$ for all $k = 1, \ldots, K$ corresponds to the observational setting.) Given a family of interventional targets $\mathcal{I} = \{I_k\}_{k=1}^K$, we built an *augmented graph* $\mathcal{G}^{\mathcal{I}}$ from the parentless context variables $\boldsymbol{C}^{\mathcal{I}}$ along with the existing system variables $\boldsymbol{X}$ by making sure $\mathrm{ch}_{\mathcal{G}}(\boldsymbol{C}_k) = I_k$. An example augmented graph with the intervention sets $\mathcal{I} = \{\emptyset, \{X_3\}, \{X_4\}\}$ can be found in Figure 8. The new system represented by the augmented graph is called *meta system*, and the structural equations governing the meta system can be written in the following way:

$$f_i'(\boldsymbol{X}_{pa_{\mathcal{G}}(i)}, \boldsymbol{C}^{\mathcal{I}}_{\mathrm{pa}^{\mathcal{I}}_{\mathcal{G}}(i)}, \epsilon_i, \tilde{\epsilon}_i) = \begin{cases} \tilde{f}_i^{(k)}(\boldsymbol{X}_{pa_{\mathcal{G}}(i)}) + \tilde{\epsilon}_i^{(k)}, & \text{if } \exists k \in [K] \text{ s.t. } \boldsymbol{C}_K \neq \emptyset \text{ and } X_i \in I_k, \\ f_i(\boldsymbol{X}_{pa_{\mathcal{G}}(i)}) + \epsilon_i, & \text{otherwise.} \end{cases} \quad (16)$$

Recall from (7) that the probability distribution for the interventional setting $I_k$ can be written as:

$$p_{I_k, \mathcal{G}}(\boldsymbol{X}) = p_E\Big([(\mathbf{id} - \mathbf{U}_k \boldsymbol{f})(\boldsymbol{X})]_{\mathcal{U}_k}\Big) \times p_{\widetilde{E}}\Big([(\mathbf{id} - \tilde{\mathbf{U}}_k \tilde{\boldsymbol{f}})(\boldsymbol{X})]_{I_k}\Big) \times \big|\det(\mathbf{J}_{(\mathbf{id} - \boldsymbol{f}^{(I_k)})}(\boldsymbol{X}))\big|.$$

This can also be written in terms of the context variables in the following way:

$$p_{\mathcal{G}^{\mathcal{I}}}(\boldsymbol{X} \mid \boldsymbol{C}_k = \boldsymbol{\xi}_{I_k}, \boldsymbol{C}_{-k} = \emptyset) = p_{I_k, \mathcal{G}}(\mathbf{X}),$$

where $\boldsymbol{\xi}_{I_k}$ is an indicator function. Moreover, the joint distribution can be expressed as:

$$p_{\mathcal{G}^{\mathcal{I}}}(\boldsymbol{C}^{\mathcal{I}}, \boldsymbol{X}) = p_{\mathcal{G}^{\mathcal{I}}}(\boldsymbol{C}^{\mathcal{I}}) p_{\mathcal{G}^{\mathcal{I}}}(\boldsymbol{X} \mid \boldsymbol{C}^{\mathcal{I}}). \quad (17)$$

**Definition B.5.** Let $\mathcal{G} = (\mathcal{V}, \mathcal{E})$ be a directed graph, and $\mathcal{I} = \{I_k\}_{k=0}^K$ with $I_0 = \emptyset$ be a family of interventional targets. Let $\mathcal{M}_{\mathcal{I}}(\mathcal{G})$ denote the set of positive densities $p_{\mathcal{G}^{\mathcal{I}}} : \mathbb{R}^d \to \mathbb{R}$ such that $p_{\mathcal{G}^{\mathcal{I}}}$ is given by (17) for all $\boldsymbol{f}^{(I_k)} : \mathbb{R}^d \to \mathbb{R}^d$, with $f_i^{(I_k)}(\boldsymbol{X}) = f_i^{(I_k)}(\boldsymbol{X}_{\mathrm{pa}_{\mathcal{G}}}(i))$, such that $\boldsymbol{f}^{(I_k)}$ is unique and invertible.

**Proposition B.6.** *For a directed graph $\mathcal{G} = (\mathcal{V}, \mathcal{E})$ and a family of interventional targets $\mathcal{I} = \{I_k\}_{k=0}^K$ such that $I_0 = \emptyset$, let $p \in \mathcal{M}_{\mathcal{I}}(\mathcal{G})$, then $p$ satisfies the general directed global Markov property relative to $\mathcal{G}^{\mathcal{I}}$.*

*Proof.* For a directed graph $\mathcal{G}$, suppose the intervened mechanisms $\boldsymbol{f}^{(I_k)}$ are uniquely specified and invertible. Then the corresponding structural equations admit a unique solution on each strongly connected component of $\mathcal{G}$. Moreover, introducing context variables in the augmented graph does not create additional cycles, so the resulting meta-system constitutes a simple SCM. Consequently, by Theorem A.21 in (Bongers et al., 2021), the induced distribution $p_{\mathcal{G}^{\mathcal{I}}}$ is well-defined (unique) and satisfies the general directed global Markov property. $\qquad\square$

We now introduce the interventional Markov equivalence class for directed graphs, defined in terms of the set of distributions they induce.

**Definition B.7** ($\mathcal{I}$-Markov Equivalence Class). Two directed graphs $\mathcal{G}_1$ and $\mathcal{G}_2$ are $\mathcal{I}$-Markov equivalent if and only if $\mathcal{M}_{\mathcal{I}}(\mathcal{G}_1) = \mathcal{M}_{\mathcal{I}}(\mathcal{G}_2)$, denoted as $\mathcal{G}_1 \equiv_{\mathcal{I}} \mathcal{G}_2$. The set of all directed graphs that are $\mathcal{I}$-Markov equivalent to $\mathcal{G}_1$ is the $\mathcal{I}$-Markov equivalence class of $\mathcal{G}_1$, denoted as $\mathcal{I}$-MEC($\mathcal{G}_1$).

### B.3. Proof of Theorem 3.1

In this section, we prove the main theorem of this paper. We recall the score function introduced in Section 3.4:

$$\mathcal{S}(\mathcal{G}, \mathcal{I}) = \sup_{\boldsymbol{\theta}} \sum_{k=1}^{K} \mathbb{E}_{\boldsymbol{X} \sim p^{(k)}} \big[ \log p_{I_k, \mathcal{G}}(\boldsymbol{X}) \big] - \lambda_{\mathcal{G}} |\mathcal{G}| - \lambda_{\mathcal{I}} |\mathcal{I}|,$$

where $p^{(k)}$ is the ground truth distribution for the $k$-th experiment, and $\boldsymbol{\theta}$ is the parameters of the causal system. We can rewrite the score function for the metasystem introduced above as:

$$\mathcal{S}(\mathcal{G}^{\mathcal{I}}) = \sup_{\boldsymbol{\theta}} \mathbb{E}_{(\boldsymbol{X}, \boldsymbol{C}) \sim p_{\mathcal{I}}^*} \log p_{\mathcal{G}^{\mathcal{I}}}(\boldsymbol{X}, \boldsymbol{C} \mid \boldsymbol{\theta}) - \lambda' |\mathcal{G}^{\mathcal{I}}|,$$

where $p_{\mathcal{I}}^*$ is the joint ground truth distribution for all of the variables in the augmented graph and the $p_{\mathcal{G}^{\mathcal{I}}}(\boldsymbol{X}, \boldsymbol{C} \mid \boldsymbol{\theta})$ is given by (17) for a specific choice of $\boldsymbol{\theta}$. We define $\mathcal{P}_{\mathcal{I}}(\mathcal{G})$ as the collection of all distributions $p_{\mathcal{G}^{\mathcal{I}}}(\boldsymbol{X}, \boldsymbol{C} \mid \boldsymbol{\theta})$ that can be represented by the model specified in (5), (8), and (9). That is,

$$\mathcal{P}_{\mathcal{I}}(\mathcal{G}) := \{p \mid \exists \boldsymbol{\theta} \text{ s.t } p = p_{\mathcal{G}^{\mathcal{I}}}(\cdot \mid \boldsymbol{\theta})\}. \tag{18}$$

Theorem 3.1 relies on four assumptions. First of which is to ensure that the model is able to express the ground truth distribution.

**Assumption B.8** (Sufficient Capacity). The joint ground truth distribution $p_{\mathcal{I}}^*$ is such that $p_{\mathcal{I}}^* \in \mathcal{P}_{\mathcal{I}}^*(\mathcal{G}^*)$, where $\mathcal{G}^*$ is the ground truth graph and $\mathcal{I}^*$ is the ground truth intervention family.

The second assumption is the generalization of the faithfulness to the interventional setting.

**Assumption B.9** ($\mathcal{I}$-$\sigma$-faithfulness). Let $\boldsymbol{V} = (\boldsymbol{X}, \boldsymbol{C}^{\mathcal{I}})$, for any subset of nodes $A, B, C \subseteq \mathcal{V} \cup \boldsymbol{C}^{\mathcal{I}}$, and $I_k \in \mathcal{I}$

$$A \overset{\sigma}{\underset{\mathcal{G}^{\mathcal{I}}}{\not\perp}} B \mid C \implies \boldsymbol{V}_A \underset{p_{\mathcal{G}^{\mathcal{I}}}}{\not\perp} \boldsymbol{V}_B \mid \boldsymbol{V}_C.$$

The above assumption entails that any conditional independence observed in the data must correspond to a $\sigma$-separation in the associated interventional ground-truth graph. Third assumption is to ensure the model distribution is strictly positive.

**Assumption B.10** (Strict positivity). The joint model distribution $p_{\mathcal{G}^{\mathcal{I}}}(\cdot \mid \boldsymbol{\theta})$ is strictly positive for all parameters $\boldsymbol{\theta}$, directed graph $\mathcal{G}$ and interventional family $\mathcal{I}$.

From Assumption B.10 and (18) we can see that $\mathcal{P}_{\mathcal{I}}(\mathcal{G}) \subseteq \mathcal{M}_{\mathcal{I}}(\mathcal{G})$.

**Assumption B.11** (Finite differential entropy). For $\mathcal{I} = \{I_k\}_{k=0}^{K}$,

$$|\mathbb{E}_{p_{\mathcal{I}}^*} \log p_{\mathcal{I}}^*(\boldsymbol{X}, \boldsymbol{C})| < \infty.$$

The final assumption is to ensure that both $\mathcal{S}(\mathcal{G}^{\mathcal{I}})$ and $\mathcal{S}(\mathcal{G}^{*\mathcal{I}^*})$ don't go to infinity, as illustrated by the following lemma from (Brouillard et al., 2020).

**Lemma B.12** (Finiteness of the score function (Brouillard et al., 2020)). *Under assumptions B.8 and B.11, $\mathcal{S}(\mathcal{G}^{\mathcal{I}}) < \infty$.*

Using the results of (Brouillard et al., 2020), we can write the score difference between $\mathcal{G}^{*\mathcal{I}^*}$ and $\mathcal{G}^{\mathcal{I}}$ as a KL-divergence minimization term plus the difference between their regularization penalties.

**Lemma B.13** (Rewritting the score function (Brouillard et al., 2020))**.** *Under assumptions B.8 and B.11, we have*

$$\mathcal{S}(\mathcal{G}^{*\mathcal{I}^*}) - \mathcal{S}(\mathcal{G}^{\mathcal{I}}) = \inf_{\theta} D_{KL}(p_{\mathcal{I}}^* \| p_{\mathcal{G}^{\mathcal{I}}}(\cdot \mid \boldsymbol{\theta})) + \lambda'(|\mathcal{G}^{\mathcal{I}}| - |\mathcal{G}^{*\mathcal{I}^*}|).$$

In order to prove Theorem 3.1 we will take the following technical lemma from (Sethuraman & Fekri, 2025).

**Lemma B.14.** *Let $\mathcal{G} = (\mathcal{V}, \mathcal{E})$ be a directed graph, for a set of interventional targets $\mathcal{I} = \{I_k\}_{k=0}^{K}$, and $p^* \notin \mathcal{M}_{\mathcal{I}}(\mathcal{G}))$, then*

$$\inf_{p \in \mathcal{M}_{\mathcal{I}}(\mathcal{G}))} D(p^* \| p) > 0.$$

The proof of this lemma can be found in (Sethuraman & Fekri, 2025).

We recall Theorem 3.1 and present its proof.

**Theorem B.15.** *3.1 Let $\mathcal{G}^*$ be the ground truth graph and let $\mathcal{I}^*$ be the ground truth intervention family. $(\hat{\mathcal{G}}, \hat{\mathcal{I}}) \in \arg\max_{\mathcal{G},\mathcal{I}} \mathcal{S}(\mathcal{G}, \mathcal{I})$. Under the Assumptions 2.1, B.8, B.9, B.10 and B.11, and for suitable $\lambda_{\mathcal{G}}, \lambda_{\mathcal{I}} > 0$, $\hat{\mathcal{G}}$ is $\mathcal{I}^*$-Markov equivalent to $\mathcal{G}^*$ and $\hat{\mathcal{I}} = \mathcal{I}^*$.*

*Proof.* We should show that if $\mathcal{G} \notin \mathcal{I}$-MEC$(\mathcal{G}^*)$ or $\mathcal{I} \neq \mathcal{I}^*$, the score function for the augmented graph $\mathcal{G}^{\mathcal{I}}$ will be strictly lower than the score function of $\mathcal{G}^{*\mathcal{I}^*}$, i.e $\mathcal{S}(\mathcal{G}^{\mathcal{I}}) < \mathcal{S}(\mathcal{G}^{*\mathcal{I}^*})$. To show that, first, we define

$$\eta(\mathcal{G}^{\mathcal{I}}) := \inf_{\theta} D_{KL}(p_{\mathcal{I}}^* \| p_{\mathcal{G}^{\mathcal{I}}}(\cdot \mid \boldsymbol{\theta})).$$

Then, from Lemma B.13, the difference between these two score functions can be written as

$$\mathcal{S}(\mathcal{G}^{*\mathcal{I}^*}) - \mathcal{S}(\mathcal{G}^{\mathcal{I}}) = \eta(\mathcal{G}^{\mathcal{I}}) + \lambda'(|\mathcal{G}^{\mathcal{I}}| - |\mathcal{G}^{*\mathcal{I}^*}|) \tag{19}$$

Since $\mathcal{G} \notin \mathcal{I}$-MEC$(\mathcal{G}^*)$ or $\mathcal{I} \neq \mathcal{I}^*$, $\mathcal{G}^{*\mathcal{I}^*}$ and $\mathcal{G}^{\mathcal{I}}$ do not impose the same $\sigma$-separation constraints; that means there must exist subsets of nodes $A, B, C \subseteq \mathcal{V} \cup \boldsymbol{C}^{\mathcal{I}}$ such that either:

$$A \underset{\mathcal{G}^{\mathcal{I}}}{\overset{\sigma}{\perp}} B \mid C \quad \text{and} \quad A \underset{\mathcal{G}^{*\mathcal{I}^*}}{\overset{\sigma}{\not\perp}} B \mid C,$$

or

$$A \underset{\mathcal{G}^{\mathcal{I}}}{\overset{\sigma}{\not\perp}} B \mid C \quad \text{and} \quad A \underset{\mathcal{G}^{*\mathcal{I}^*}}{\overset{\sigma}{\perp}} B \mid C,$$

From Assumption B.8 we know $p_{\mathcal{I}}^* \in \mathcal{M}_{\mathcal{I}}(\mathcal{G}^*)$, then for the first case it must be true that $\boldsymbol{V}_A \not\perp_{p^{(k)}} \boldsymbol{V}_B \mid \boldsymbol{V}_C$ (Assumption B.9). Therefore, $p_{\mathcal{I}}^*$ doesn't satisfy the general directed Markov property with respect to $\mathcal{G}^{\mathcal{I}}$ and hence $p_{\mathcal{I}}^* \notin \mathcal{M}_{\mathcal{I}}(\mathcal{G})$. For the second case if we take $p_{\mathcal{I}}^* \in \mathcal{M}_{\mathcal{I}}(\mathcal{G})$ then from Assumption B.9 we can say $\boldsymbol{V}_A \not\perp_{p^{(k)}} \boldsymbol{V}_B \mid \boldsymbol{V}_C$, however since $p_{\mathcal{I}}^* \in \mathcal{M}_{\mathcal{I}}(\mathcal{G}^*)$ and Proposition B.6 implies that $\boldsymbol{V}_A \perp_{p^{(k)}} \boldsymbol{V}_B \mid \boldsymbol{V}_C$, this is a contradiction. Therefore, $p_{\mathcal{I}}^* \notin \mathcal{M}_{\mathcal{I}}(\mathcal{G})$. Thus, by applying Lemma B.14, we can show $\eta(\mathcal{G}^{\mathcal{I}})$ would be strictly positive. This would imply the score difference in (19) would be always positive for the scenarios where $|\mathcal{G}^{\mathcal{I}}| \geq |\mathcal{G}^{*\mathcal{I}^*}|$. By picking $\lambda'$ such that $0 < \lambda' < \min_{\mathcal{G}^{\mathcal{I}} \in \mathbb{G}^+} \frac{\eta(\mathcal{G}^{\mathcal{I}})}{|\mathcal{G}^{*\mathcal{I}^*}| - |\mathcal{G}^{\mathcal{I}}|}$, we can make sure (19) would remain positive for $\mathcal{G}^{\mathcal{I}} \in \mathbb{G}^+ := \{\mathcal{G}^{\mathcal{I}} \mid |\mathcal{G}^{\mathcal{I}}| < |\mathcal{G}^{*\mathcal{I}^*}|\}$. We can see this from

$$\lambda' < \min_{\mathcal{G}^{\mathcal{I}} \in \mathbb{G}^+} \frac{\eta(\mathcal{G}^{\mathcal{I}})}{|\mathcal{G}^{*\mathcal{I}^*}| - |\mathcal{G}^{\mathcal{I}}|} \tag{20}$$

$$\iff \lambda < \frac{\eta(\mathcal{G}^{\mathcal{I}})}{|\mathcal{G}^{*\mathcal{I}^*}| - |\mathcal{G}^{\mathcal{I}}|} \quad \forall \mathcal{G}^{\mathcal{I}} \in \mathbb{G}^+ \tag{21}$$

$$\iff \lambda(|\mathcal{G}^{*\mathcal{I}^*}| - |\mathcal{G}^{\mathcal{I}}|) < \eta(\mathcal{G}^{\mathcal{I}}) \quad \forall \mathcal{G}^{\mathcal{I}} \in \mathbb{G}^+ \tag{22}$$

$$\iff 0 < \eta(\mathcal{G}^{\mathcal{I}}) + \lambda(|\mathcal{G}^{\mathcal{I}}| - |\mathcal{G}^{*\mathcal{I}^*}|) = \mathcal{S}(\mathcal{G}^{*\mathcal{I}^*}) - \mathcal{S}(\mathcal{G}^{\mathcal{I}}) \quad \forall \mathcal{G}^{\mathcal{I}} \in \mathbb{G}^+. \tag{23}$$

Therefore, we have shown that for every graph that is outside of the general directed Markov equivalence class of the ground truth graph, and every interventional family different from the ground truth interventional family, would yield a strictly lower score. □

## B.4. Characterization of Equivalence Class

A graphical notion of the $\mathcal{I}$-Markov equivalence class of a direct graph $\mathcal{G}$ can be given using $\sigma$-Maximal Ancestral Graphs ($\sigma$-MAGs) (Yao & Mooij, 2025). A graph $\mathcal{G}$ is said to be *maximal* if there exists no inducing path (relative to the empty set) between any two non-adjacent nodes. An *inducing path* relative to a subset $L$ is a path on which every non-endpoint node $i \notin L$ is a collider on the path, and every collider is an ancestor of an endpoint of the path. A *Maximal Ancestral Graph* (MAG) is one that is both ancestral and maximal. A $\sigma$-*MAG* for a directed graph $\mathcal{G}$ is a MAG on the same node set that represents the $\sigma$-separation model of $\mathcal{G}$ in the sense that $\sigma$-separation in $\mathcal{G}$ coincides with $m$-separation (defined as in (Yao & Mooij, 2025)) in the $\sigma$-MAG. Given the augmented graph $\mathcal{G}^{\mathcal{I}}$, it is possible to construct a $\sigma$-MAG over $\boldsymbol{V} = (\boldsymbol{X}, \boldsymbol{C}^{\mathcal{I}})$ that preserves both the independence structure and ancestral relationships encoded in $\mathcal{G}^{\mathcal{I}}$; see (Yao & Mooij, 2025) for details. We denote $\sigma$-MAG($\mathcal{G}^{\mathcal{I}}$) to mean a $\sigma$-MAG constructed from $\mathcal{G}^{\mathcal{I}}$. Therefore, all independencies encoded by $\sigma$-separation in $\mathcal{G}^{\mathcal{I}}$ are also present in $\sigma$-MAG($\mathcal{G}^{\mathcal{I}}$) via $m$-separation. A path $\pi = (i_0, \varepsilon_1, \ldots, i_{n-1}, \varepsilon_n, i_n)$ in $\sigma$-MAG($\mathcal{G}^{\mathcal{I}}$) is called a *discriminating path* for $i_{n-1}$ if (1) $\pi$ includes at least three edges; (2) $i_{n-1}$ is a non-endpoint node on $\pi$, and is adjacent to $i_n$ on $\pi$; and (3) $i_0$ and $i_n$ are not adjacent, and every node in between $i_0$ and $i_{n-1}$ is a collider on $\pi$ and is a parent of $i_n$. The following theorem characterizes the equivalence of $\sigma$-MAGs.

**Theorem B.16** ((Yao & Mooij, 2025)). *Two $\sigma$-MAGs $\mathcal{G}_1$ and $\mathcal{G}_2$ are Markov equivalent if and only if:*

1. *$\mathcal{G}_1$ and $\mathcal{G}_2$ have the same adjacencies;*

2. *$\mathcal{G}_1$ and $\mathcal{G}_2$ have the same unshielded colliders; and*

3. *Let $\pi$ be a discriminating path for a node $v$ in $\mathcal{G}_1$, and let $\pi'$ be the corresponding path to $\pi$ in $\mathcal{G}_2$ If $\pi'$ is also a discriminating path for $v$, then $v$ is a collider on $\pi$ in $\mathcal{G}_1$ if and only if it is a collider on $\pi'$ in $\mathcal{G}_2$.*

Hence, by Theorem B.16, two directed graphs $\mathcal{G}_1$ and $\mathcal{G}_2$ are $\mathcal{I}$-Markov equivalent if and only if their corresponding $\sigma$-MAGs, $\sigma$-MAG($\mathcal{G}_1^{\mathcal{I}}$) and $\sigma$-MAG($\mathcal{G}_2^{\mathcal{I}}$), satisfy the conditions of Theorem B.16; that is, (i) have the same skeleton, (ii) have the same unshielded colliders, and (iii) have the same discriminating paths with consistent collider status.

# C. Additional Experiments

## C.1. Experiments on Non-contractive DAGs

We conduct tests on non-contractive causal mechanisms where the ground truth graph is acyclic. We modify our methodology to work under non-contractive SEM's following the preconditioning approach proposed by (Sethuraman et al., 2023). According to this method we introduce a learnable diagonal preconditioning matrix $\boldsymbol{\Lambda}$ to transform the causal mechanism in the following way:

$$\hat{\boldsymbol{f}} = \boldsymbol{\Lambda}^{-1} \circ \boldsymbol{f} \circ \boldsymbol{\Lambda}, \tag{24}$$

where $\boldsymbol{f}$ remains contractive. For this comparison, we additionally include DCDI (Brouillard et al., 2020), UT-IGSP (Squires et al., 2020), and BACADI (Hägele et al., 2022) as baseline methods. These approaches are designed for learning DAGs under unknown interventions. A direct comparison with UT-IGSP is infeasible, since, as a constraint-based approach, it does not return a candidate graph but instead a candidate I-Markov Equivalence class. To evaluate the AUPRC, we picked the maximum among the graphs in this equivalence class. Figure 9 shows that SCOUT can recover the causal structure of the graph with a near-perfect performance in every setting except the Noisy Function + Gaussian noise scenario, where it obtains comparable results with baselines. As for the target recovery, Table 4 suggests SCOUT successfully identifies the intervened nodes for shift and scale interventions.

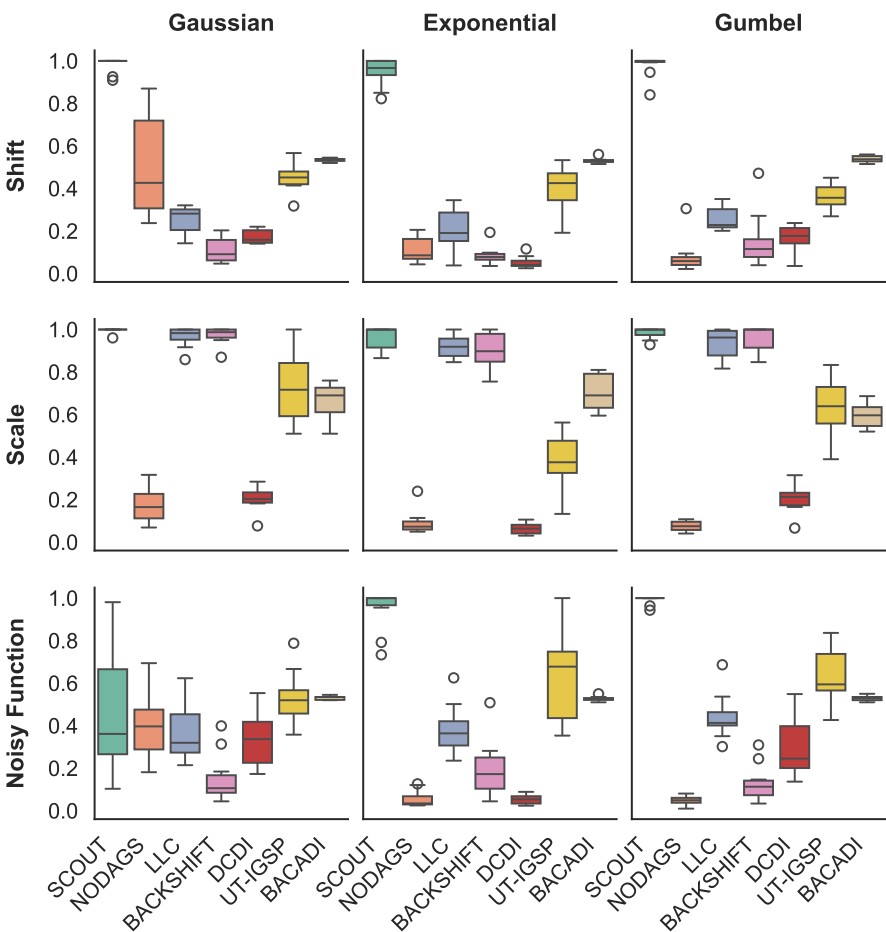

*Figure 9.* Graph recovery performance comparison between SCOUT and baselines under non-contractive DAG's and various interventional and exogenous noise settings, evaluated using AUPRC. In all cases, the number of nodes is fixed at $d = 10$.

*Table 4.* Graph recovery performance comparison between SCOUT and baselines under different intervention types and exogenous noise distributions, evaluated using AUPRC (higher is better). Results are reported as mean $\pm$ standard deviation.

| Noise Type | Intervention Type | SCOUT | BACKSHIFT | DCDI | UT-IGSP | BACADI |
|---|---|---|---|---|---|---|
| Gaussian | Shift | $1.000 \pm 0.000$ | $0.954 \pm 0.054$ | $0.157 \pm 0.215$ | $0.991 \pm 0.019$ | $0.887 \pm 0.032$ |
| | Scale | $1.000 \pm 0.000$ | $1.000 \pm 0.000$ | $0.104 \pm 0.140$ | $1.000 \pm 0.000$ | $0.766 \pm 0.019$ |
| | Noisy Function | $0.137 \pm 0.063$ | $0.166 \pm 0.127$ | $0.204 \pm 0.239$ | $0.212 \pm 0.108$ | $0.550 \pm 0.000$ |
| Exponential | Shift | $1.000 \pm 0.000$ | $0.979 \pm 0.030$ | $0.158 \pm 0.193$ | $0.924 \pm 0.041$ | $1.000 \pm 0.000$ |
| | Scale | $1.000 \pm 0.000$ | $1.000 \pm 0.000$ | $0.056 \pm 0.000$ | $0.909 \pm 0.064$ | $1.000 \pm 0.000$ |
| | Noisy Function | $0.322 \pm 0.171$ | $0.145 \pm 0.102$ | $0.098 \pm 0.047$ | $0.366 \pm 0.152$ | $0.550 \pm 0.000$ |
| Gumbel | Shift | $1.000 \pm 0.000$ | $0.825 \pm 0.183$ | $0.055 \pm 0.000$ | $0.983 \pm 0.030$ | $0.825 \pm 0.033$ |
| | Scale | $1.000 \pm 0.000$ | $1.000 \pm 0.000$ | $0.057 \pm 0.001$ | $0.995 \pm 0.014$ | $0.870 \pm 0.033$ |
| | Noisy Function | $0.158 \pm 0.062$ | $0.131 \pm 0.073$ | $0.109 \pm 0.157$ | $0.292 \pm 0.105$ | $0.550 \pm 0.000$ |

## C.2. Experiments for Linear SEM

We evaluate SCOUT's performance alongside baselines for linear SEM, using the same intervention and noise settings as in the nonlinear case. We use AUPRC as our evaluation metric (higher is better) again. The box plot results of Figure 10 show that SCOUT can again achieve near-perfect graph recovery in all settings except Noisy Function + Gaussian noise (where it outperforms all of the baselines). From Table 5, it can be seen that SCOUT can also achieve near-perfect intervention target recovery, except for noisy function interventions, where it still outperforms BACKSHIFT.

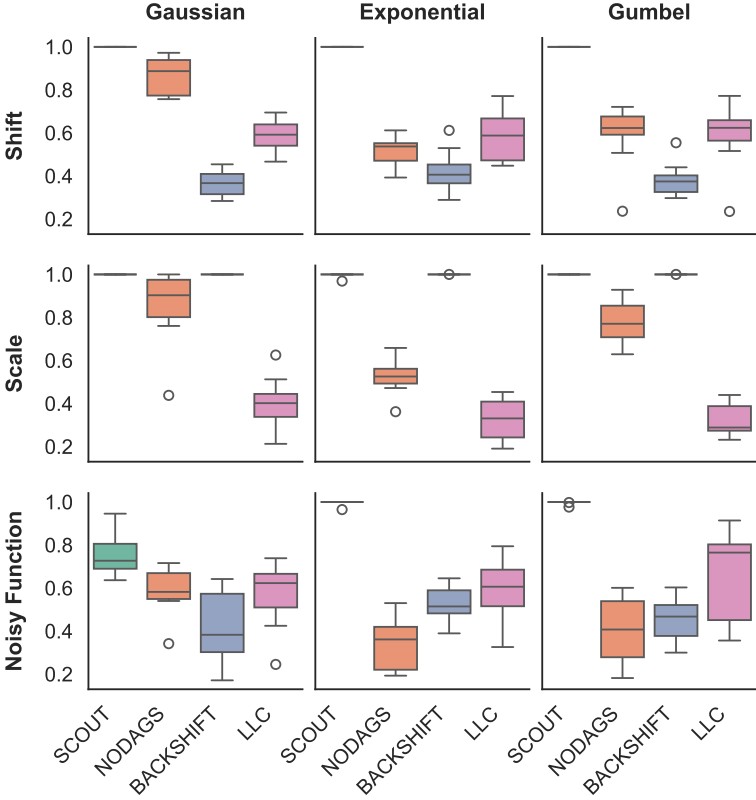

*Figure 10.* Graph recovery performance comparison between SCOUT and baselines under linear SEM and various interventional and exogenous noise settings, evaluated using AUPRC. In all cases, the number of nodes is fixed at $d = 10$.

*Table 5.* Interventional target recovery performance comparison between SCOUT and BACKSHIFT under linear SEM across different intervention modes and exogenous noise types. The table presents the average AUPRC value along with its standard deviation evaluated using 10 independent trials.

| Intervention Type | Gaussian | | Exponential | | Gumbel | |
|---|---|---|---|---|---|---|
| | SCOUT | BACKSHIFT | SCOUT | BACKSHIFT | SCOUT | BACKSHIFT |
| Shift | $1.000 \pm 0.000$ | $0.812 \pm 0.173$ | $1.000 \pm 0.000$ | $0.816 \pm 0.182$ | $1.000 \pm 0.000$ | $0.707 \pm 0.186$ |
| Scale | $1.000 \pm 0.000$ | $1.000 \pm 0.000$ | $0.956 \pm 0.138$ | $1.000 \pm 0.000$ | $1.000 \pm 0.000$ | $1.000 \pm 0.000$ |
| Noisy Function | $0.591 \pm 0.169$ | $0.404 \pm 0.304$ | $0.842 \pm 0.189$ | $0.533 \pm 0.170$ | $0.792 \pm 0.080$ | $0.445 \pm 0.251$ |

## C.3. Experiments Under Known Intervention Targets

We conduct a performance benchmark of graph recovery for SCOUT and baselines under known interventions for nonlinear SEM (The BACKSHIFT model does not accept known targets, so we use the unknown setting for this baseline). Figure 11 shows that SCOUT can recover the graph with near-perfect performance across all settings, including Noisy Function + Gaussian noise, indicating that the distributional shift in this setting allows our model to learn the graph structure when the targets are known.

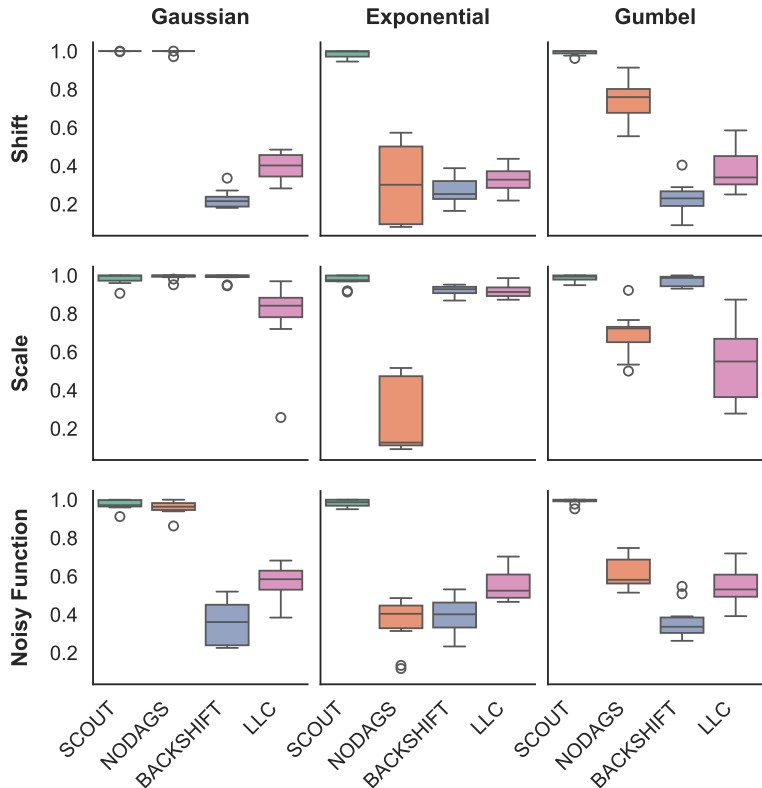

*Figure 11.* Graph recovery performance comparison between SCOUT and baselines for known intervention targets under nonlinear SEM and various interventional and exogenous noise settings, evaluated using AUPRC. In all cases, the number of nodes is fixed at $d = 10$.

## C.4. Experiments for Hard (Perfect) Interventions

We run experiments on SCOUT and the baselines to evaluate their structures and target recovery performance under hard (perfect) interventions, in which the incoming edges of the intervened nodes are removed. Figure 12 shows that SCOUT can learn causal relationships under hard interventions. Both SCOUT and BACKSHIFT can identify intervention targets in this setting with perfect accuracy.

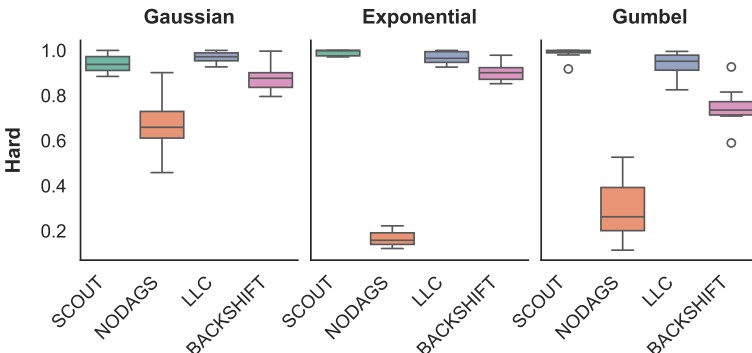

*Figure 12.* Graph recovery performance comparison between SCOUT and baselines under nonlinear SEM and hard interventions with various exogenous noise settings, evaluated using AUPRC. In all cases, the number of nodes is fixed at $d = 10$.

### C.5. Ablation Studies

#### C.5.1. IMPACT OF NUMBER OF MAXIMUM INTERVENTIONAL TARGETS

In this section, we evaluated SCOUT's performance alongside baselines while varying the maximum number of interventions per experiment from 1 to 5. We examine non-linear SEM under unknown-scale interventions with Gaussian and Gumbel noise. From Figure 13 and Table 6, we observe that as the maximum number of intervention targets per experiment increases, SCOUT shows some performance degradation. Nevertheless, it consistently remains superior to the baseline methods. In contrast, BACKSHIFT deteriorates much more substantially in both graph structure recovery and intervention target recovery compared to the single-node intervention setting.

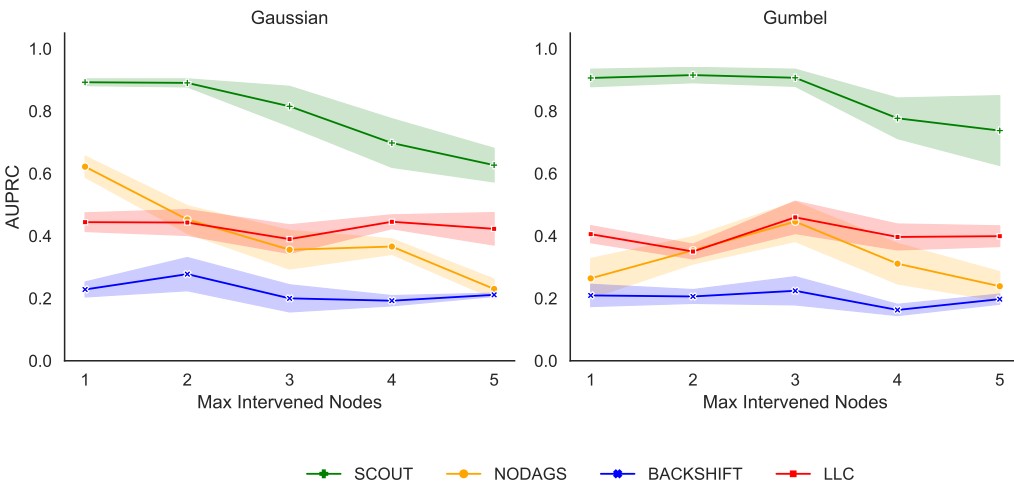

*Figure 13.* Graph recovery performance comparison between SCOUT and baselines under non-linear SEM and shift interventions. The number of maximum intervention targets per experiment is varied from 1 to 5.

*Table 6.* Interventional target recovery performance comparison between SCOUT and BACKSHIFT under non-linear SEM and shift interventions. The table presents the average AUPRC value along with its standard deviation evaluated using 10 independent trials. The number of nodes is varied from 1 to 5.

| Max Intervened Nodes | Gaussian | | Gumbel | |
| --- | --- | --- | --- | --- |
| | SCOUT | BACKSHIFT | SCOUT | BACKSHIFT |
| 1 | $1.000 \pm 0.000$ | $0.641 \pm 0.158$ | $1.000 \pm 0.000$ | $0.451 \pm 0.155$ |
| 2 | $1.000 \pm 0.000$ | $0.372 \pm 0.165$ | $0.990 \pm 0.023$ | $0.435 \pm 0.188$ |
| 3 | $0.986 \pm 0.031$ | $0.401 \pm 0.128$ | $0.999 \pm 0.001$ | $0.471 \pm 0.048$ |
| 4 | $0.940 \pm 0.053$ | $0.298 \pm 0.077$ | $0.912 \pm 0.066$ | $0.300 \pm 0.077$ |
| 5 | $0.849 \pm 0.104$ | $0.356 \pm 0.080$ | $0.919 \pm 0.087$ | $0.321 \pm 0.124$ |

### C.5.2. SCALING WITH TRAINING SAMPLES

In this experiment, we evaluate the sample size requirements of SCOUT. Figure 14 and Table 7 indicate that even 250 samples per experiment allows SCOUT to learn the causal graph along with unknown targets under shift interventions.

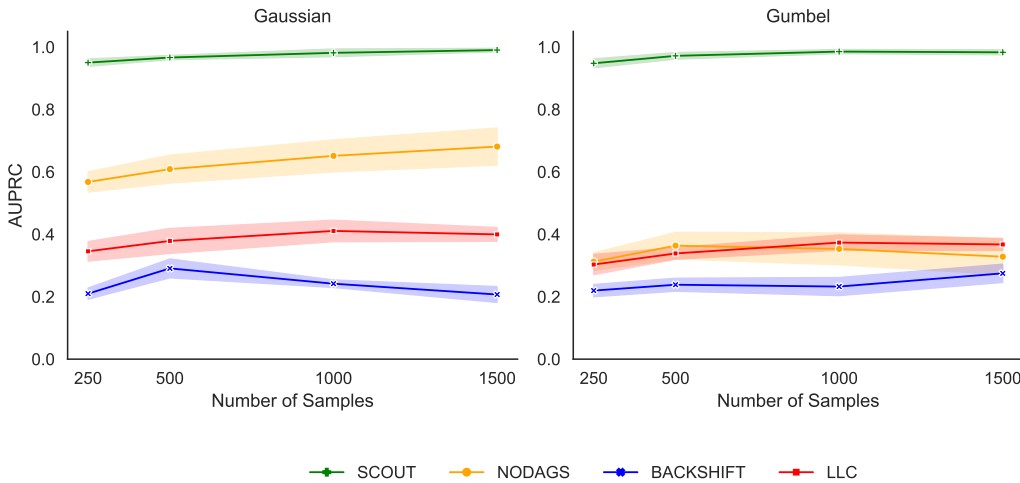

*Figure 14.* Graph recovery performance comparison between SCOUT and baselines under non-linear SEM and shift interventions. The number of samples per experiment is varied from 250 to 1500.

*Table 7.* Interventional target recovery performance comparison between SCOUT and BACKSHIFT under non-linear SEM and shift interventions. The table presents the average AUPRC value along with its standard deviation evaluated using 10 independent trials. The number of samples per experiment is varied from $n = 250$ to $n = 1500$.

| Number of Samples | Gaussian | | Gumbel | |
| --- | --- | --- | --- | --- |
| | SCOUT | BACKSHIFT | SCOUT | BACKSHIFT |
| 250 | $1.00 \pm 0.00$ | $0.82 \pm 0.18$ | $1.00 \pm 0.00$ | $0.70 \pm 0.20$ |
| 500 | $1.00 \pm 0.00$ | $0.87 \pm 0.17$ | $1.00 \pm 0.00$ | $0.79 \pm 0.16$ |
| 1000 | $1.00 \pm 0.00$ | $0.93 \pm 0.08$ | $1.00 \pm 0.00$ | $0.87 \pm 0.16$ |
| 1500 | $1.00 \pm 0.00$ | $0.92 \pm 0.18$ | $1.00 \pm 0.00$ | $0.88 \pm 0.13$ |

### C.5.3. SCALING WITH OUTGOING EDGE DENSITY

We evaluate the effect of the graph sparsity on the structure and target recovery of SCOUT by varying the expected outgoing edge density from 1 to 4. The results are summarized in Figure 15 and Table 8. The SCOUT learns the underlying graph structure and unknown interventional targets independently of graph sparsity.

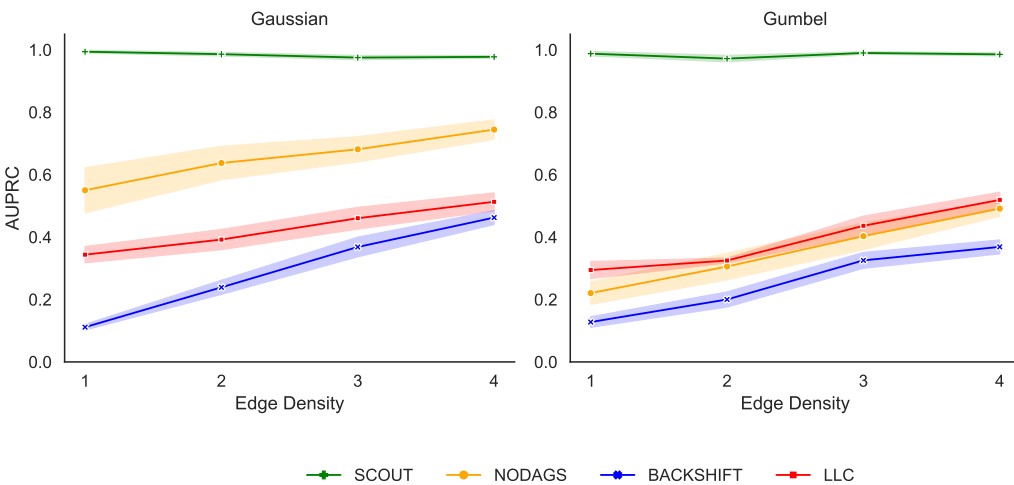

*Figure 15.* Graph recovery performance comparison between SCOUT and baselines under non-linear SEM and shift interventions. The number of expected outgoing edge density is varied from 1 to 4.

*Table 8.* Interventional target recovery performance comparison between SCOUT and BACKSHIFT under non-linear SEM and shift interventions. The table presents the average AUPRC value along with its standard deviation evaluated using 10 independent trials. Edge density is varied from 1 to 4.

|  | Gaussian | | Gumbel | |
| --- | --- | --- | --- | --- |
| Edge Density | SCOUT | BACKSHIFT | SCOUT | BACKSHIFT |
| 1 | $1.00 \pm 0.00$ | $0.98 \pm 0.04$ | $1.00 \pm 0.00$ | $0.80 \pm 0.20$ |
| 2 | $1.00 \pm 0.00$ | $0.89 \pm 0.15$ | $1.00 \pm 0.00$ | $0.82 \pm 0.13$ |
| 3 | $1.00 \pm 0.00$ | $0.94 \pm 0.10$ | $1.00 \pm 0.00$ | $0.89 \pm 0.12$ |
| 4 | $1.00 \pm 0.00$ | $0.88 \pm 0.15$ | $1.00 \pm 0.00$ | $0.82 \pm 0.17$ |

### C.5.4. IMPACT OF SHIFT PARAMETER

We evaluate the effect of the shift parameter on the graph and target recovery in this study. We vary the shift amount from 0 (observational case) to 2. From Figure 16, it can be understood that for the Gaussian noise, with the increasing shift amount, SCOUT's performance on discovering the causal relationships increases, whereas for the Gumbel noise model, it works with near-perfect performance for all shift parameters. In terms of intervention target recovery, Table 9 indicates that for Gaussian noise, SCOUT can learn targets even for low amounts of shift, and for Gumbel noise, its performance gets better as the shift parameter increases.

*Table 9.* Interventional target recovery performance comparison between SCOUT and BACKSHIFT under non-linear SEM and shift interventions. The table presents the average AUPRC value along with its standard deviation evaluated using 10 independent trials. The shift parameter is varied from 0.5 to 2.0.

|  | Gaussian | | Gumbel | |
| --- | --- | --- | --- | --- |
| Shift | SCOUT | BACKSHIFT | SCOUT | BACKSHIFT |
| 0.5 | $0.99 \pm 0.02$ | $0.24 \pm 0.11$ | $0.18 \pm 0.05$ | $0.13 \pm 0.08$ |
| 1.0 | $1.00 \pm 0.00$ | $0.63 \pm 0.21$ | $1.00 \pm 0.00$ | $0.40 \pm 0.17$ |
| 2.0 | $1.00 \pm 0.00$ | $0.95 \pm 0.05$ | $1.00 \pm 0.00$ | $0.93 \pm 0.10$ |

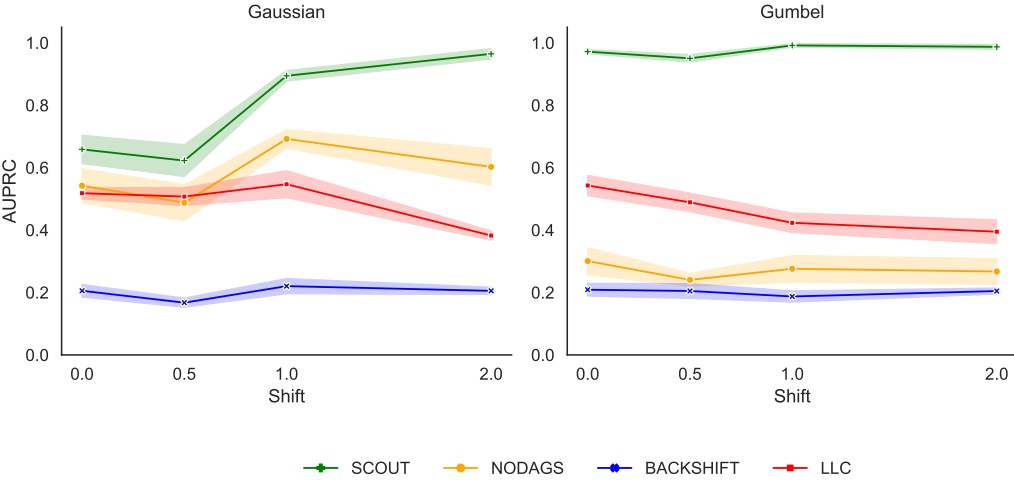

*Figure 16.* Graph recovery performance comparison between SCOUT and baselines under non-linear SEM and shift interventions. The shift parameter is varied from 0 (observational case) to 2.

### C.5.5. IMPACT OF SCALE PARAMETER

In this section, we vary the scale parameter to see its effect on the performance of SCOUT and baselines. We change it from 0.25 to 2 (0.5 is the observational case). The results are given in Figure 17 and Table 10. For Gaussian noise, the overall performance of SCOUT again increases, and for Gumbel noise, it achieves near-perfect performance regardless of the scale parameter.

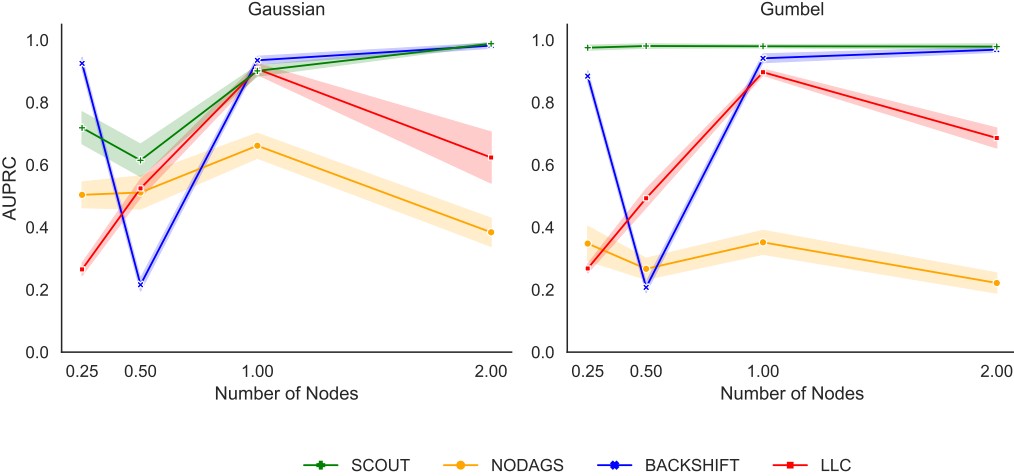

*Figure 17.* Graph recovery performance comparison between SCOUT and baselines under non-linear SEM and shift interventions. The scale parameter is varied from 0.25 to 2.

*Table 10.* Interventional target recovery performance comparison between SCOUT and BACKSHIFT under non-linear SEM and scale interventions. The table presents the average AUPRC value along with its standard deviation evaluated using 10 independent trials. The scale factor is varied from 0.25 to 2.0.

| | Gaussian | | Gumbel | |
|---|---|---|---|---|
| Scale | SCOUT | BACKSHIFT | SCOUT | BACKSHIFT |
| 0.25 | $0.05 \pm 0.00$ | $0.00 \pm 0.00$ | $0.05 \pm 0.00$ | $0.00 \pm 0.00$ |
| 1.00 | $1.00 \pm 0.00$ | $1.00 \pm 0.00$ | $1.00 \pm 0.00$ | $1.00 \pm 0.00$ |
| 2.00 | $1.00 \pm 0.00$ | $1.00 \pm 0.00$ | $1.00 \pm 0.00$ | $1.00 \pm 0.00$ |

### C.5.6. IMPACT OF CYCLES

In this section, we change the number of cycles in ground truth graph to see its effect on the performance of SCOUT and baselines. The number of nodes in the graph are fixed to $d = 10$, and the number of cycles are varied from 0 to 8. The results are given in Figure 18 and Table 11. The number of cycles do no effect the performance of SCOUT.

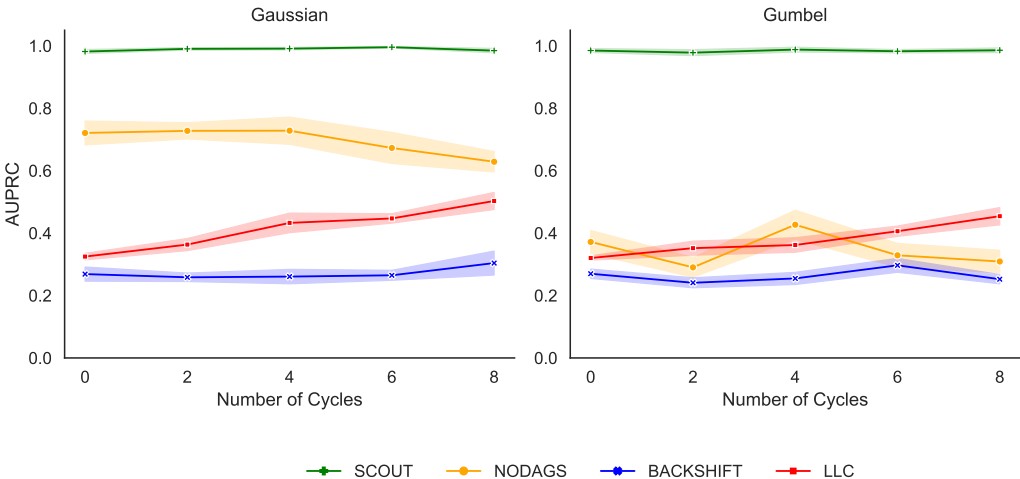

*Figure 18.* Graph recovery performance comparison between SCOUT and baselines under non-linear SEM and shift interventions. The number of cycles are varied from 0 to 8

*Table 11.* Interventional target recovery performance comparison between SCOUT and BACKSHIFT under non-linear SEM and shift interventions. The table presents the average AUPRC value along with its standard deviation evaluated using 10 independent trials. The number of cycles is varied from 0 to 8.

| | Gaussian | | Gumbel | |
|---|---|---|---|---|
| Number of Cycles | SCOUT | BACKSHIFT | SCOUT | BACKSHIFT |
| 0 | $1.00 \pm 0.00$ | $0.91 \pm 0.12$ | $1.00 \pm 0.00$ | $0.86 \pm 0.12$ |
| 2 | $1.00 \pm 0.00$ | $0.88 \pm 0.19$ | $1.00 \pm 0.00$ | $0.88 \pm 0.15$ |
| 4 | $1.00 \pm 0.00$ | $0.95 \pm 0.05$ | $1.00 \pm 0.00$ | $0.92 \pm 0.10$ |
| 6 | $1.00 \pm 0.00$ | $0.93 \pm 0.11$ | $1.00 \pm 0.00$ | $0.90 \pm 0.11$ |
| 8 | $1.00 \pm 0.00$ | $0.95 \pm 0.10$ | $1.00 \pm 0.00$ | $0.94 \pm 0.08$ |

## C.6. Induced Distribution Change Comparison of Soft Interventions

The results presented in Section 4 show that neither SCOUT nor the baselines can perfectly recover the graph structure and the intervention targets for noisy function interventions, as opposed to shift and scale interventions. However, for the known interventions, we see in Figure 11 that SCOUT can identify the causal structure with near-perfect performance. The reason is that, to recover the unknown intervention targets under the finite-data limit, we need a significant distributional change induced by that intervention (Gamella & Heinze-Deml, 2020). To experimentally verify this, we compare the KL-divergence between the single-node interventional and observational distributions under these three types of soft interventions for a graph with $d = 10$ nodes. For every node $i$ and experiment $k$, we estimate the marginal distributions of $X_i$ under intervention $(p_{k,i})$ and observational $(q_i)$ using a shared histogram binning, and compute the divergence.

$$D_{i,j} = \mathrm{KL}(p_{i,j} \,\|\, q_j) \,.$$

We then summarize the overall intervention impact by averaging across interventions and nodes:

$$\bar{D} = \frac{1}{d} \sum_{i=1}^{d} \left( \frac{1}{\mathrm{K}} \sum_{k=1}^{K} D_{k,i} \right),$$

and report the mean and standard deviation of $\bar{D}$ across the 10 runs. We can see from Figure 19 that noisy function interventions yield a significantly low KL-divergence, thus inducing a limited distribution change compared to shift/scale interventions.

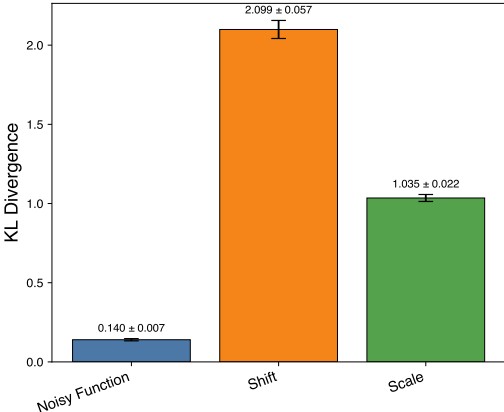

*Figure 19.* KL-divergence comparison between noisy function, shift, and scale interventions under all the single node interventions for a graph with $d = 10$ nodes.

## C.7. Additional Experiments on Perturb-CITE-seq Dataset

We test how well SCOUT performs compared to other baselines when the intervention targets are known on the Perturb-CITE-seq dataset (Frangieh et al., 2021). Additionally, we compared the baselines' performances (this time including BACKSHIFT) using *Mean Absolute Error* (MAE) as the evaluation metric. We can compute MAE by taking the mean of $\|\boldsymbol{f}^{(I_k)}(\boldsymbol{x}) - \boldsymbol{x}\|_1 / d$ over all observations x in the held-out test set. The results for both known and unknown settings are given in Figures 21 and 22, respectively. The results indicate that SCOUT remains competitive with state-of-the-art methods under the MAE metric.

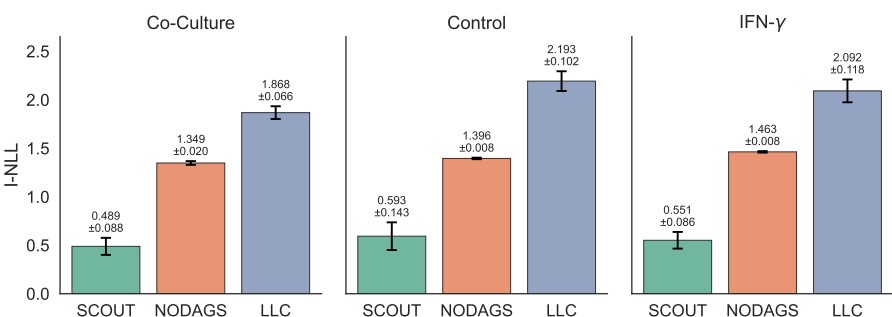

*Figure 20.* The performance comparison results on the Perturb-CITE-seq (Frangieh et al., 2021) gene perturbation dataset with known interventional targets. The bar graph shows mean absolute error (MAE) and its standard deviation across 5 trials.

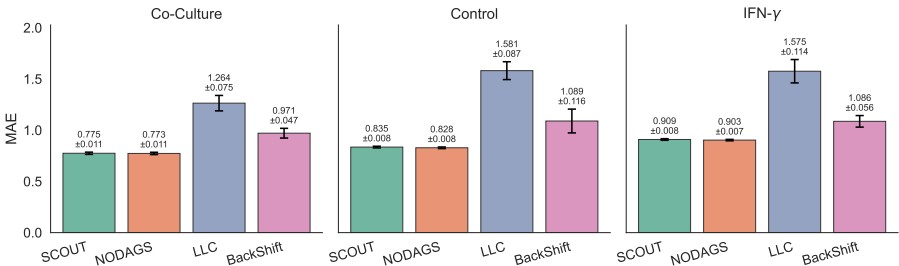

*Figure 21.* The performance comparison results on the Perturb-CITE-seq (Frangieh et al., 2021) gene perturbation dataset with unknown interventional targets. The bar graph shows mean absolute error (MAE) and its standard deviation across 5 trials.

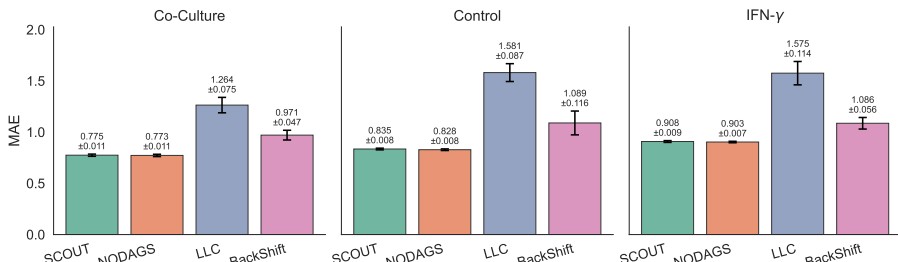

*Figure 22.* The performance comparison results on the Perturb-CITE-seq (Frangieh et al., 2021) gene perturbation dataset with unknown interventional targets. The bar graph shows interventional negative log-likelihood (I-NLL) and its standard deviation across 5 trials.

## D. Experimental Setup

In this section, we explain how we generate our synthetic data and how we preprocess the gene perturbation dataset. We provide the implementation details for SCOUT, along with the baselines, and discuss our evaluation metrics. The code for SCOUT is available at the repository: https://github.com/alparturkoglu/scout-master

### D.1. Data Generation for Synthetic Experiments

For all types of SEM, we first sample a directed graph with edge density of 2 using the Erdős-Rényi (ER) random graph model. For the linear SEM, we sample the edge weights from the uniform distribution $\text{Unif}((-0.9, 0.2) \cup (0.2, 0.9))$ for contractive SEMs used in cyclic graphs, and we rescale the edge weight matrix to ensure its Lipschitz constant is less than 1. For the nonlinear SEM, we use a single-layer MLP with tanh (rectified linear unit) activation, $\boldsymbol{f} = tanh(\boldsymbol{W}^T \boldsymbol{x})$, where $\boldsymbol{W}$ is the weighted adjacency matrix. We ensure the contractivity by rescaling with the operator norm. For the noisy function interventions, we generate the intervened causal mechanism $\tilde{\boldsymbol{f}}$ by negating the signs of the weights of the last layer for $\boldsymbol{f}$. We can see from Proposition D.1 that this approach preserves the contractivity of the combined intervened causal mechanism $\boldsymbol{f}^{(I_k)}$, thus satisfying Assumption 2.1.

**Proposition D.1.** *For an interventional experiments $I_k \in \mathcal{I}$ and a contractive causal mechanism $\boldsymbol{f}$, Let $\tilde{\boldsymbol{f}} = \alpha \boldsymbol{f}$, for*

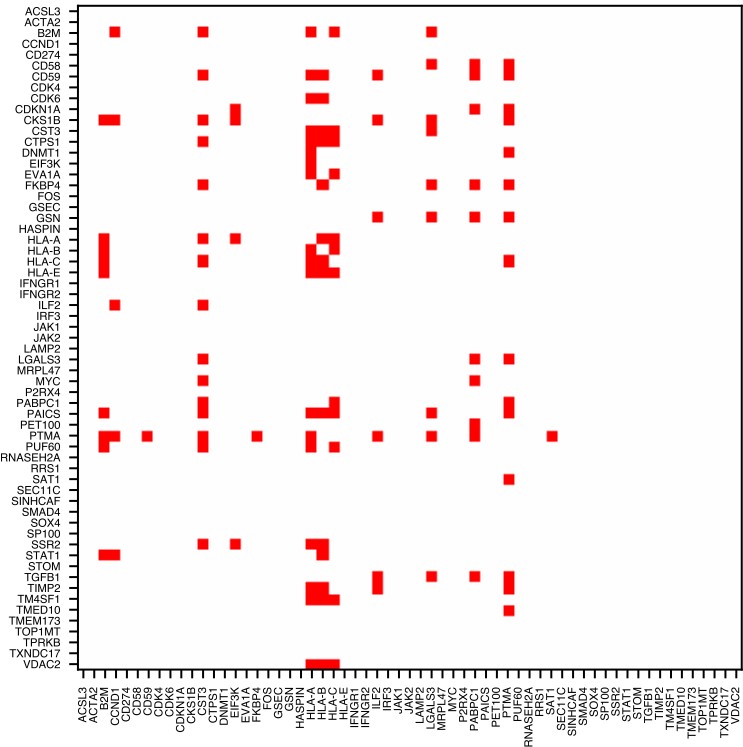

*Figure 23.* The adjacency matrix learnt by SCOUT for co-culture cell condition of Perturb-CITE-seq dataset (Frangieh et al., 2021).

$|\alpha| \leq 1$. *Then, the combined interventional causal mechanism* $\boldsymbol{f}^{(I_k)} \triangleq (\mathbf{U}_k \boldsymbol{f} + (\mathbf{1}_d - \mathbf{U}_k)\tilde{\boldsymbol{f}})$ *remains contractive.*

*Proof.* We should show that $\boldsymbol{f}^{(I_k)} \triangleq (\mathbf{U}_k \boldsymbol{f} + (\mathbf{1}_d - \mathbf{U}_k)\tilde{\boldsymbol{f}})$ will still be contractive if $\boldsymbol{f}$ is contractive and $\tilde{\boldsymbol{f}} = \alpha \boldsymbol{f}$ where $|\alpha| \leq 1$.

$$\boldsymbol{f}^{(I_k)}(\mathbf{x}) = \big(\mathbf{U}_k + \alpha(\mathbf{I} - \mathbf{U}_k)\big)\boldsymbol{f}(\mathbf{x}) = \mathbf{A}\,\boldsymbol{f}(\mathbf{x}),$$

where $\mathbf{A} := \mathbf{U}_k + \alpha(\mathbf{I} - \mathbf{U}_k)$ is diagonal with entries $a_i \in \{1, \alpha\}$. For any $\mathbf{x}, \mathbf{y} \in \mathbb{R}^d$,

$$\|\boldsymbol{f}^{(I_k)}(\mathbf{x}) - \boldsymbol{f}^{(I_k)}(\mathbf{y})\| = \|\mathbf{A}\big(\boldsymbol{f}(\mathbf{x}) - \boldsymbol{f}(\mathbf{y})\big)\|$$

$$\leq \|\mathbf{A}\|\|\boldsymbol{f}(\mathbf{x}) - \boldsymbol{f}(\mathbf{y})\|,$$

where $\|\mathbf{A}\|$ denotes the induced operator norm. Since $\mathbf{A}$ is diagonal,

$$\|\mathbf{A}\| = \max_i |a_i| = \max\{1, |\alpha|\} = 1 \quad \text{(because } |\alpha| \leq 1\text{)}.$$

Thus,

$$\|\boldsymbol{f}^{(I_k)}(\mathbf{x}) - \boldsymbol{f}^{(I_k)}(\mathbf{y})\| \leq \|\boldsymbol{f}(\mathbf{x}) - \boldsymbol{f}(\mathbf{y})\| \leq L\,\|\mathbf{x} - \mathbf{y}\|,$$

which shows that $\boldsymbol{f}^{(I_k)}$ is contractive with Lipschitz constant at most $L < 1$.

Therefore, $\boldsymbol{f}^{(I_k)}$ is contractive. $\qquad \square$

### D.2. Gene Perturbation Dataset

The data were downloaded from the Broad Institute Single Cell Portal (accession SCP1064). Following the preprocessing protocol of (Sethuraman et al., 2023), we removed cells with fewer than 500 detected genes and discarded genes expressed in fewer than 500 cells. To keep the analysis computationally tractable, we restricted attention to 61 perturbed genes (Table 12) selected from the full set of measured genes. We then split the data by experimental condition (co-culture, IFN-$\gamma$, and control), training and evaluating models separately within each condition. .

*Table 12.* The selected gene set from the Perturb-CITE-seq dataset (Frangieh et al., 2021).

| ACSL3 | ACTA2 | B2M | CCND1 | CD274 | CD58 | CD59 | CDK4 | CDK6 |
|-------|-------|-----|-------|-------|------|------|------|------|
| CDKN1A | CKS1B | CST3 | CTPS1 | DNMT1 | EIF3K | EVA1A | FKBP4 | FOS |
| GSEC | GSN | HASPIN | HLA-A | HLA-B | HLA-C | HLA-E | IFNGR1 | IFNGR2 |
| ILF2 | IRF3 | JAK1 | JAK2 | LAMP2 | LGALS3 | MRPL47 | MYC | P2RX4 |
| PABPC1 | PAICS | PET100 | PTMA | PUF60 | RNASEH2A | RRS1 | SAT1 | SEC11C |
| SINHCAF | SMAD4 | SOX4 | SP100 | SSR2 | STAT1 | STOM | TGFB1 | TIMP2 |
| TM4SF1 | TMED10 | TMEM173 | TOP1MT | TPRKB | TXNDC17 | VDAC2 | | |

## D.3. Implementation Details

### D.3.1. ARCHITECTURAL DETAILS

The main algorithm of SCOUT is implemented in Python using the *PyTorch* library. The model takes as input an interventional dataset, where each sample is associated with its corresponding experiment index, along with a set of hyperparameters.

To model the causal mechanisms $f$ and $\tilde{f}$, we use a `gumbelSoftMLP` architecture without hidden layers, followed by a `tanh` activation. A learned Gumbel-sigmoid adjacency mask is used to select the parent set of each node, while intervention targets are inferred using a separate Gumbel-sigmoid intervention mask.

We do not employ temperature annealing; instead, the temperatures are fixed at $1.0$ for graph structure learning and $0.5$ for intervention-target learning. The algorithm is initialized with parameters $\theta^{(0)}, M_\phi^{(0)}, T_\psi^{(0)}$, and proceeds according to the methodology described in Section 3 to maximize the proposed score function.

The objective is optimized using the ADAM optimizer (Kingma & Ba, 2017). The hyperparameters used during training are reported in Table 13, and the overall training procedure is summarized in Algorithm 1. All experiments were conducted on NVIDIA RTX6000 GPUs.

---

**Algorithm 1** SCOUT Training

---

**Require:** Interventional dataset $\{x^{(i)}\}_{i=1}^N$, experiment index $\{k^{(i)}\}_{i=1}^N$, regularization coefficients $\lambda_\mathcal{G}$ and $\lambda_\mathcal{I}$, batch size $B$, learning rate $\alpha$.

**Ensure:** Learned neural network parameters $\hat{\theta}$, graph structure parameters $\hat{M}_\phi$, interventional target parameters $\hat{T}_\psi$.

1: Initialize the parameters: $\theta^{(0)} \sim p_\theta(\theta)$, $M_\phi^{(0)} \sim p_{M_\phi}(M_\phi)$, and $T_\psi^{(0)} \sim p_{T_\psi}(T_\psi)$
2: **while NOT** CONVERGED **do**
3:     Shuffle $\{x^{(i)}, k^{(i)}\}_i$
4:     **for** $t = 1$ to $N/B$ **do**
5:         $M^{(t)} \sim M_\phi$
6:         $T^{(t)} \sim T_\psi$
7:         Compute LOSS $= -\frac{1}{B}\sum_{i=1}^B \hat{\mathcal{S}}\big(\theta, \{x^{(j)}\}_{j=B(t-1)}^{Bt}, \{k^{(j)}\}_{j=B(t-1)}^{Bt}, M^{(t)}, T^{(t)}\big) + \lambda_\mathcal{G}\|M_\phi\|_1 + \lambda_\mathcal{I}\|T_\psi\|_1$
8:         Backpropagate using ADAM(LOSS, $M_\phi, T_\phi, \theta, \alpha$)
9:         Perform RESCALE($f_\theta, \tilde{f}_\theta$) to ensure $f_\theta$ and $\tilde{f}_\theta$ are 0.9-Lipschitz
10:     **end for**
11: **end while**
12: **return** $\hat{\theta}, \hat{M}_\phi, \hat{T}_\psi$

---

*Table 13.* Hyperparameters used in our experiments.

| Hyperparameter | Meaning | Value |
|---|---|---|
| $\lambda_{\mathcal{G}}$ | Graph sparsity regularizer | 0.001 |
| $\lambda_{\mathcal{I}}$ | Intervention family sparsity regularizer | 0.01 |
| $\alpha$ | Learning rate | 0.01 |
| $B$ | Batch size | 512 |

### D.3.2. SENSITIVITY ANALYSIS

To assess SCOUT's sensitivity to random seeds and initialization, we conducted the following experiment. Using a fixed ground-truth graph and intervention targets with shift interventions under Gaussian noise and a non-linear mechanism, we trained the model five times with different random initializations. The graph recovery AUPRC achieves a mean of 0.9854 with a standard deviation of 0.00439, while the intervention recovery AUPRC is 1.0 in all trials. These results indicate that SCOUT is robust to initialization and random seed variability.

To evaluate sensitivity to hyperparameters, we trained SCOUT using a range of hyperparameter settings on the same graph and intervention targets. As shown in Table 14, SCOUT achieves near-perfect performance across a wide range of configurations, demonstrating robustness to hyperparameter choices.

To analyze the variance of the log-determinant estimator, we considered a linear SEM setting where the true determinant can be computed analytically. The estimator achieves a variance of $3.297 \times 10^{-3}$ and a mean squared error (MSE) of $2.905 \times 10^{-1}$.

*Table 14.* SCOUT performance for various choices of hyperparameters $\alpha$, $\lambda_c$, $\lambda_r$

| Learning rate ($\alpha$) | $\lambda_c$ | $\lambda_r$ | AUPRC | Int. AUPRC |
|---|---|---|---|---|
| $10^{-2}$ | $10^{-3}$ | $10^{-2}$ | 1.00 | 1.00 |
| $10^{-1}$ | $10^{-3}$ | $10^{-2}$ | 0.95 | 1.00 |
| $10^{-3}$ | $10^{-3}$ | $10^{-2}$ | 1.00 | 1.00 |
| $10^{-2}$ | $10^{-3}$ | $10^{-3}$ | 1.00 | 0.92 |
| $10^{-2}$ | $10^{-3}$ | $10^{-1}$ | 1.00 | 1.00 |
| $10^{-2}$ | $10^{-2}$ | $10^{-2}$ | 0.90 | 1.00 |
| $10^{-2}$ | $10^{-4}$ | $10^{-2}$ | 1.00 | 1.00 |

### D.3.3. COMPUTATIONAL COST ANALYSIS

Figure 24 reports the training times of SCOUT and the baseline methods. In contrast to the gradient-based approaches, LLC and BACKSHIFT require no stochastic optimization; as a result, they are substantially faster. NODAGS-Flow has lower runtime than SCOUT, but its formulation does not support unknown-target estimation or neural spline flows for exogenous noise transformation. All runtimes are measured on graphs with $d = 10$ nodes, using training data with all single-node interventional datasets; SCOUT and NODAGS are trained for 200 epochs.

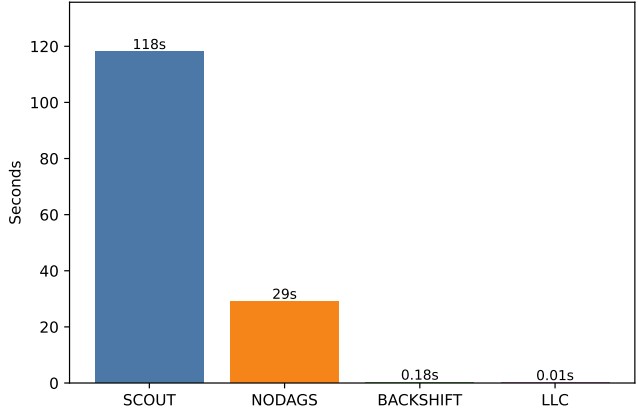

*Figure 24.* Training time comparison between SCOUT and the baselines.

Let $N$ be the number of nodes, $B$ the minibatch size, $M$ the number of samples scored at likelihood time, and $K$ the number of power-series terms in the residual log-det estimator. In the current setup, $\mathbb{E}[K] = 4$.

For one training step, the cost can be written explicitly as

$$
T_{\text{train}} = \underbrace{\mathcal{O}(BN^2)}_{\text{sample Gumbel adjacency mask}} + \underbrace{\mathcal{O}(BN^3)}_{\text{compute } f(x)} + \underbrace{\mathcal{O}(BN^3)}_{\text{compute } f_i(x)}
$$
$$
+ \underbrace{\mathcal{O}((K+1)BN^3)}_{\text{Neumann-series residual log-det}} + \underbrace{\mathcal{O}(BN)}_{\text{1D spline-flow log-det}} + \underbrace{\mathcal{O}((K+1)BN^3)}_{\text{backpropagation}}
$$
$$
+ \underbrace{\mathcal{O}(N^2)}_{\text{optimizer \& Lipschitz projection}}.
$$

For one likelihood evaluation on $M$ samples, the cost is

$$
T_{\text{lik}} = \underbrace{\mathcal{O}(MN^2)}_{\text{sample Gumbel adjacency mask}} + \underbrace{\mathcal{O}(MN^3)}_{\text{compute } f(x)} + \underbrace{\mathcal{O}(MN^3)}_{\text{compute } f_i(x)}
$$
$$
+ \underbrace{\mathcal{O}(KMN^3)}_{\text{power-series residual log-det}} + \underbrace{\mathcal{O}(MN)}_{\text{1D spline-flow log-det}}.
$$

Hence, both training and likelihood evaluation are dominated by cubic scaling in the number of nodes, with overall leading-order costs $\mathcal{O}((K+1)BN^3)$ and $\mathcal{O}(KMN^3)$, respectively.

We have measured the per-epoch computation time as well as training memory footprint for SCOUT as a function of nodes and experiments. Tables 15 and 16 indicate that the model scales linearly with the number of experiments where as the main computational bottleneck is the scaling with the number of nodes which is expected because of the Jacobian calculation.

*Table 15.* Per-epoch computation time and training memory as a function of the number of nodes.

| Nodes | Experiments | Time / epoch (s) | Train-State Mem Est. (KiB) |
|---|---|---|---|
| 10 | 10 | 0.59 | 21.7656 |
| 30 | 30 | 8.30 | 102.7031 |
| 50 | 50 | 75.00 | 233.6406 |
| 70 | 70 | 197.0 | 414.5781 |

*Table 16.* Per-epoch computation time and training memory as a function of the number of experiments.

| Nodes | Experiments | Time / epoch (s) | Train-State Mem Est. (KiB) |
|---|---|---|---|
| 10 | 10 | 0.59 | 21.7656 |
| 10 | 20 | 1.20 | 23.3281 |
| 10 | 30 | 1.80 | 24.8906 |

### D.3.4. BASELINES

For NODAGS-Flow, we used the authors' public implementation (Sethuraman et al., 2023) and kept all hyperparameters at their default values. We implemented LLC following the description in Hyttinen et al. (2012); our implementation is provided in the baselines folder of the supplementary materials. For DCDI and BACKSHIFT, we used the official author-provided codebases available at https://github.com/slachapelle/dcdi and https://github.com/christinaheinze/backShift, respectively. For BACADI, we used the public repository at https://github.com/haeggee/bacadi. UT-IGSP was run using the causaldag Python package, and SERGIO simulations were generated using the official SERGIO repository at https://github.com/PayamDiba/SERGIO.

### D.4. Evaluation Metrics

We use the Area Under Precision-Recall Curve (AUPRC) as our general evaluation metric. AUPRC computes the area under the precision-recall curve evaluated at various threshold values (the higher the better).

$$\text{Precision} = \frac{TP}{TP + FP}, \qquad \text{Recall} = \frac{TP}{TP + FN}$$

where $TP$, $FP$, and $FN$ denote true positives, false positives, and false negatives, respectively.

### D.4.1. GRAPH PROXY FOR NONLINEAR SEM VIA SQUARED JACOBIAN

For nonlinear SEMs the ground-truth causal graph is not explicitly available in the form of a linear weight matrix. To obtain a reference adjacency for evaluation, we calculate a squared Jacobian proxy for the causal mechanism $\boldsymbol{f}$, We estimate the squared Jacobian entries

$$S_{ij} \triangleq \mathbb{E}_{\mathbf{x} \sim \mathcal{X}} \left[ \left( \frac{\partial f_i(\mathbf{x})}{\partial x_j} \right)^2 \right], \tag{25}$$

where $\mathcal{X}$ is a chosen set of probe inputs. In practice, we approximate (25) empirically using automatic differentiation and averaging over a finite set of sampled points:

$$\widehat{S}_{ij} = \frac{1}{N} \sum_{n=1}^{N} \left( \frac{\partial f_i(\mathbf{x}^{(n)})}{\partial x_j} \right)^2. \tag{26}$$

We threshold the sensitivity matrix to obtain a binary adjacency:

$$\widehat{A}_{ij} = \mathbb{I} \left[ \widehat{S}_{ij} > \tau \right], \tag{27}$$

with threshold $\tau = 0.001$ in our experiments.

