# OpenReview forum: "SCOUT: Cyclic Causal Discovery Under Soft Interventions with Unknown Targets"
_ICML.cc/2026/Conference — ICML 2026 regular_

### Official Review · Reviewer_vQmt · 2026-02-18

**Soundness:** 2
**Presentation:** 3
**Significance:** 2
**Originality:** 3
**Overall Recommendation:** 4
**Confidence:** 4

**Summary:**

This paper studies learning directed (possibly cyclic) graphs under multiple intervention contexts with unknown targets. The method relies on flexible density estimation using normalizing flows, and present a procedure for selecting a graph and intervention family by optimizing a penalized score. Experiments are conducted on synthetic and real-world datasets.

**Compliance With Llm Reviewing Policy:**

Affirmed.

**Final Justification:**

My major concerns on "The interventions in cyclic settings are characterized in an over simplified way" and "The equilibrium is assumed for granted" have been partly addressed. Though flaws still exist in conflating the use of d- and sigma- separations, I believe that can be addressed with certain (relatively suitable) assumptions. I have thus raised my score to 4.

**Key Questions For Authors:**

see "weaknesses".

**Limitations:**

see "weaknesses".

**Strengths And Weaknesses:**

## Strengths:

1. The topic is relevant and important. The interventional cyclic causal discovery in nonlinear settings seems largely unexplored.
2. The technical development about score estimation is formal and standard, though I didn't check the correctness in detail.

## Weaknesses:

My major concern is that the paper focuses heavily on the estimation machinery (e.g., density models) but **leaves the basic identifiability and causal modeling of interventions in cyclic systems largely undeveloped.** Here are some details.

1. **The interventions in cyclic settings are characterized in an over simplified way.**
   - The authors directly take interventions on targets as changing their incoming causal mechanisms, and let the non-targets' causal mechanisms remain unchanged.
   - This is only standard in acyclic setting. However, with cycles, except for the changing mechanisms on the targets, the whole equilibrium state of the dynamic system may also be altered, leading to changing mechanisms even on non-targets (e.g., http://proceedings.mlr.press/r5/dash05a/dash05a.pdf, https://arxiv.org/pdf/2205.10083).
   - Hence, more subtle characterizations on the intervention itself, or at least justifications for assuming interventions in the proposed way, are needed.

2. **The equilibrium is assumed for granted, while in practice there may be much more subtleties.**
   - In this work, the authors directly assume that X is a snapshot of a dynamic system under equilibrium (under Equation 2).
   - However, this may depend on the graph structure itself. A graph may have none or multiple parameterizations that can reach an equilibrium. Such characterization has to be made carefully.
   - Also, Assumption 2.1 is named "Interventional stability". This can be incorrect, since the invertibility does not imply the stability in any sense; think of linear settings, stability is related more on the convergence, instead of the invertibility, of the related matrices.
   - Overall, similar to point 1, much more discussions are needed for when an intervention can reach an equilibrium, and how.

3. **The identification objective of the "I-Markov equivalence class" needs more discussion and justification.**
   - In Theorem 3.1, the authors directly take the I-Markov equivalence class (augmenting the original causal graph with intervention indicators pointing to corresponding targets) as the identification objective.
   - First of all, the meanings of G* and I* seem undefined before using them.
   - In Appendix A, the authors directly use the sigma-separations as the implications for conditional independencies and the conditional density invariances observed in data. While this is correct for the nonlinear cyclic part among X, for the CI between intervention indicators I and X, it may need more justification on why sigma-separations are still complete.
   - Especially under the shift interventions / scale interventions settings, they look like linear interventions, and one may naturally concern that there are more conditional density invariances other than those implied by sigma-separations (instead, by original d-separations).
   - Also, the interventional equivalence class is only defined but not characterized. It remains unclear when two graphs, under interventions and nonlinear mechanisms, are Markov equivalent. It also remains unclear which graphical rules (like Meek rules in acyclic setting) can be used to completely traverse/represent the whole MEC under prior knowledge (all edges adjacent to interventions are outgoing).
   - Overall, more discussions on characterization, and running examples of this MEC identification objective, are expected.

4. **Do soft interventions differ essentially from hard interventions in cyclic, nonlinear setting? Similarly, does the "unknown targets" impose substantial technical difficulties in this setting?**
   - In acyclic setting, it is known that soft and hard interventions do not differ essentially for the purpose of causal discovery; by using the conditional density invariance constraints, the true equivalence class can always be characterized and discovered in a same principled way.
   - Similarly, the "unknown targets" is also not a difficulty; true targets can be identified as those whose conditional densities given any other variables are always changing across domains.
   - It would be valuable to discuss whether such nice and simple properties still hold in this paper's setting. At least for the second one, I guess that "unknown targets" is not a technical difficulty either.

5. **What is the intention for nonlinear causal mechanisms and non-Gaussian noise components?**
   - It would be better to separate identifiability assumptions needed for recovering the target equivalence class, and the estimation assumptions needed for consistent scoring.
   - And state which components are "must have", or "for flexibility".

6. **Why only three intervention families?**
   - Currently, only three specific types of soft interventions are considered: shift, scale, and noisy function interventions.
   - It remains unclear why the interventions are restricted the way it is, and whether the method can generalize to arbitrary mechanism and noise changes.

---

> ### Author Rebuttal · Authors · 2026-03-30
>
> We thank the reviewer for their thorough and constructive feedback. We truly appreciate the recognition of our findings for cyclic causal discovery in nonlinear settings as relevant and important. We address the reviewer's concerns about the modeling of interventions and identifiability, and clarify our modeling assumptions.
> ## Regarding W1 and W2
> We thank the reviewer for pointing out these concerns regarding our intervention characterization. The dynamical system that we assume in the observational case can be written as
> $$
> x^{(t)} := f(x^{(t-1)}) + \\varepsilon.
> $$
> If $f$ is contractive, then by Banach's fixed-point theorem, for every initial value $x^{(0)}$ and every $\\varepsilon$, the sequence $\\{x^{(t)}\\}_{t \\ge 0}$ converges to a unique fixed point $x^{\\star}$ satisfying
> $$
> x^{\\star} = f(x^{\\star}) + \\varepsilon.
> $$
> Similarly, for an intervention that changes the mechanism or the noise characteristics, the dynamical system can be written as
>
> $$
> x^{(t)} = U_k (f(x^{(t-1)}) + \\varepsilon) + (I-U_k)(f_{\\mathrm{int}}(x^{(t-1)}) + \\varepsilon_{\\mathrm{int}})
> $$
>
> where $U_k$ is a diagonal matrix with $1$'s for observed variables and $0$'s for intervened variables, and $I$ is the identity matrix. Equivalently, defining the intervened mechanism and effective noise by
>
> $$
> f^{(I_k)}(x^{(t-1)}) := U_k f(x^{(t-1)}) + (I-U_k)f_{\\mathrm{int}}(x^{(t-1)}),
> $$
>
> $$
> \\eta := U_k\\varepsilon + (I-U_k) \\varepsilon_{\\mathrm{int}},
> $$
>
> we can write
>
> $$
> x^{(t)} := f^{(I_k)}(x^{(t-1)}) + \\eta.
> $$
>
> If $f^{(I_k)}$ is contractive, then again by Banach's fixed-point theorem, for every initialization, the iterates converge to a unique fixed point $x_k^{\\star}$ satisfying
>
> $$
> x_k^{\\star} = f^{(I_k)}(x_k^{\\star}) + \\eta.
> $$
>
>
> Thus, if the intervened mechanism $f^{(I_k)}$ remains contractive, Banach's fixed-point theorem implies that the intervention also admits a unique equilibrium.
>
> In Assumption 2.1, our intention was to state that, for each intervention setting considered in our work, there exists a unique equilibrium, and hence a well-defined SCM solution, rather than to assert a stronger notion of stability. We will rephrase this point and clarify the assumption in the revised manuscript.
>
> ## Regarding W3
> We thank the reviewer for their comment. $G^{\\ast}$  and $I^{\\ast}$ are respectively the ground truth causal adjacency and the actual targets for the performed experiments. We will make sure to define these terms before they appear in the paper.
>
> Regarding the dependence between the context variables and the endogenous variables. From Equation 16 of the paper, the relationships between intervention indicators C and X are non-linear, even in the case of shift/scale interventions; thus, $\sigma$-separations can be used to infer implications for conditional independencies.
>
> In cyclic graphs, the feedback loops make it difficult to find a concise graphical representation for the equivalence class. Therefore, the characterization of the equivalence class for cyclic graphs is not as straightforward as the acyclic case, nor is it as easy as finding graphical rules like Meek rules for the case of DAGs. However, several works have sought to overcome this challenge using Maximal Ancestral Graphs (MAGs) [1], [2]. Further details, including the equivalence conditions for $\sigma$-MAGs [2], can be found in the paper in Appendix A.4. (Characterization of Equivalence Class).
> ## Regarding W4
> We thank the reviewer for pointing out this aspect. Unlike DAGs, there is a distinction between soft and hard interventions in the context of cyclic graphs due to the presence of feedback loops. For soft interventions, the graph structure does not change, and any intervention on a node that is part of a cycle would feed back to itself through the existing feedback loop, affecting the conditional densities of its descendants. For hard interventions, we remove all incoming edges to the intervened node, breaking the existing cycle (feedback loop), making hard interventions indistinguishable from the DAG case. For unknown interventions, the same feedback loop effect makes it challenging to identify the intervention target, since the conditional densities of the descendants of the target node also vary.
> ## Regarding W5
> We thank the reviewer for this question. We do not assume non-Gaussianity or non-linearity for identifiability, but we use them as a generalization and a source of flexibility for our model. SCOUT works under any noise regime, including Gaussian (see Figure 1), and under any contractive structural equation model, including linear (see Figure 8).
> ## Regarding W6
> We thank the reviewer for this question. The method we proposed can be generalized to arbitrary noise changes and mechanism changes as long as the contractivity of the overall mechanism is preserved.
> - [1] Claassen et al., Establishing Markov equivalence in cyclic directed graphs, 2023.
> - [2] Yao et al., σ-maximal ancestral graphs, 2025.

---

> > ### Author Rebuttal · Reviewer_vQmt · 2026-04-02
> >
> > I thank the authors for the response. However, my concerns are not addressed.
> >
> > Regarding W1&W2 ("The interventions in cyclic settings are characterized in an over simplified way", "The equilibrium is assumed for granted, while in practice there may be much more subtleties") -- I understand the authors' point in assuming f's being contractive to reach equilibrium. What I am asking for is the **justification** for this assumption:
> > - What if the intervention changes the f's contractility, and when would this happen?
> > - What if the dynamical process has not reached an equilibrium?
> > - What if there are multiple possible equilibrium?
> > - When you write the Uk.. + (1-Uk).. equation, it seems that the intervention is applied from scratch (t=0). But this is not always the case in practice.
> > - Also, your response "In Assumption 2.1, our intention was to state that, .. there exists a unique equilibrium" seems incorrect. Invertibility only says about "uniqueness", and nothing about the "equilibrium" implicitly intended.
> >
> > Regarding W3 ("The identification objective of the "I-Markov equivalence class" needs more discussion and justification") -- why "the relationships between intervention indicators C and X are non-linear, even in the case of shift interventions"? It seems that the shift intervention can be written as X=.. + \mu * C ?
> >
> > This concern on using sigma-separation also extends to your response to W5, "SCOUT works under any noise regime, including Gaussian (see Figure 1), and under any contractive structural equation model, including linear (see Figure 8)" -- it seems that at least for linear Gaussian SEM, d-separations should be used?

---

> > > ### Author Response · Authors · 2026-04-03
> > >
> > > We thank the reviewer for their follow-up questions.
> > >
> > > ## Regarding W1 and W2
> > >
> > > Our causal discovery framework relies on existence of a uniquely
> > > observational/interventional distribution and Markov properties with
> > > respect to $\sigma$/$d$-separation. These requirements are trivially
> > > satisfied for acyclic SCMs. However, when cycles are allowed we require
> > > stronger restrictions on the SCM to guarantee unique solvability. To
> > > that end, \[1\] showed that unique solvability with respect to each
> > > strongly connected component is necessary for an SCM to satisfy Markov
> > > property with respect to $\sigma$-separation. \[1\] also characterize a
> > > class of SCMs (called simple SCMs) that obey unique solvability
> > > requirements (which remain a standard assumption in several several
> > > constraint-based causal discovery methods \[2, 3\]. This motivates our
> > > choice of restricting the causal mechanism to be contractive.
> > > Contractivity gurantees unique solvability and obey Markov property with
> > > respect to $\sigma$-separation. The SCMs under consideration in our work
> > > is a subset of simple SCMs. Additionally, contractivity assumption also
> > > provides several computation benefits while modeling the data
> > > distribution. Primarily enabling efficient Jacobian computation via
> > > power-series approximation.
> > >
> > > Under soft interventions, the SCM need not remain simple; in other
> > > words, the system may no longer have a unique interventional
> > > distribution. Thus, Markov property is no longer satisfied, and causal
> > > discovery from static data in this regime is no longer a well-posed
> > > problem. We agree that contractivity is a restrictive assumption, and
> > > that real systems may violate it, but this was a modeling choice that
> > > was made in order to balance generality with computation tractability.
> > >
> > > Given the contractivity assumption, the post-intervention system admits
> > > a unique equilibrium from any initial condition. Therefore, within this
> > > contractive regime, the precise time at which the intervention is
> > > applied, whether at initialization or after the system has already
> > > evolved, does not affect the equilibrium: once the intervention is
> > > active, the post-intervention dynamics converge to the same unique fixed
> > > point.
> > >
> > > ## Regarding W3
> > >
> > > We thank the reviewer for this clarification. We agree that, in the
> > > special case of additive shift interventions, the *local* structural
> > > dependence of the target variable on the intervention indicator can be
> > > written in an affine form, e.g.
> > > $$X_i = f_i(X_{\mathrm{pa}(i)}) + \mu_i C + \varepsilon_i.$$ Thus, our
> > > use of $\sigma$-separation is not motivated by the local edge
> > > $C \to X_i$ being nonlinear. Rather, it is motivated by the fact that
> > > the induced equilibrium dependence of downstream variables on $C$ is
> > > generally nonlinear once the target belongs to a feedback loop. For
> > > example, consider the cyclic system
> > > $$X_1 = \mu C + \tanh(X_2) + \varepsilon_1,
> > > \qquad
> > > X_2 = \tanh(X_1) + \varepsilon_2.$$ Here the direct dependence of $X_1$
> > > on the intervention indicator $C$ is affine. However, substituting the
> > > first equation into the second yields
> > > $$X_2 = \tanh\\bigl(\mu C + \tanh(X_2) + \varepsilon_1\bigr) + \varepsilon_2,$$
> > > so the induced dependence of $X_2$ on $C$ is nonlinear. Thus, even
> > > though the local intervention term is linear in $C$, the overall
> > > equilibrium dependence propagating through the feedback loop is
> > > nonlinear.
> > >
> > > ## Regarding W5
> > >
> > > We agree with the reviewer on this point. Although the main focus of the
> > > paper is the general setting of nonlinear cyclic SEMs under arbitrary
> > > noise regimes, our empirical results also show that SCOUT performs well
> > > in the linear Gaussian case (see Figure 8). We agree that this special
> > > case deserves a separate theoretical discussion using
> > > $d$-separation-based arguments, which we will add the updated
> > > manuscript.
> > >
> > > \[1\] Bongers et al., Foundations of Structural Causal Models with
> > > Cycles and Latent Variables, 2021.
> > >
> > > \[2\] Mooij et al., Constraint-Based Causal Discovery using Partial
> > > Ancestral Graphs in the presence of Cycles, 2020.
> > >
> > > \[3\] Mokhtarian et al., A Unified Experiment Design Approach for Cyclic
> > > and Acyclic Causal Models, 2023.

---

### Official Review · Reviewer_yAsG · 2026-03-10

**Soundness:** 3
**Presentation:** 3
**Significance:** 3
**Originality:** 2
**Overall Recommendation:** 4
**Confidence:** 4

**Summary:**

The paper tackles causal discovery in a setting where several assumptions commonly used by mainstream methods are violated. The causal graph may contain directed cycles, mechanisms may be nonlinear, exogenous noise may be non-Gaussian, and the available data come from multiple soft-interventional experiments whose intervention targets are unknown. The authors propose SCOUT as a likelihood-based, differentiable framework that learns the cyclic causal graph structure, the observational and interventional mechanisms, and the unknown intervention target set per experiment. They model cyclic SEM under equilibrium with an invertible forward map as $X = f(X) + \epsilon$. They assume a unique equilibrium solution for each noise term. They parameterize the invertible mapping from $X$ to noise using contractive residual-style flows. Contractiveness ensures invertability. They model non-Gaussian noise by transforming each noise component to a standard Gaussian using an invertible CDF flow. Both the adjacency mask and the intervention-target matrix are learned via Gumbel-Softmax estimators with sparsity regularizers. In the infinite-data limit, exact maximization of their score recovers the interventional Markov equivalence class and recovers the true intervention family under the assumptions.

**Compliance With Llm Reviewing Policy:**

Affirmed.

**Final Justification:**

I have clarified the justification in the rebuttal acknowledgement.

**Key Questions For Authors:**

My major concerns are listed in the weakness section.

1. Most synthetic experiments and runtime reporting appear at $d=10$. Can you provide more results and runtime scaling for larger graphs and varying number of experiments or samples, or clarify the main computational bottleneck?

2. You cite several unknown-target methods like UT-IGSP, BaCaDI, and sparse-change approaches. Which of these are truly inapplicable to your settings, and can you add comparisons in the DAG subcases or explain why comparisons are not meaningful?

3. Since SCOUT jointly learns graph structure, intervention targets, and nonlinear mechanisms, how sensitive are results to random seeds, initialization, and hyperparameters?

4. Could you provide an explicit complexity breakdown per training step and per likelihood evaluation?

**Limitations:**

I do not see any discussion on limitations.

**Strengths And Weaknesses:**

Strengths:

1. The paper tackles a combination of cyclic graphs + nonlinear mechanisms + non-Gaussian noise + soft interventions with unknown targets that is rarely handled end-to-end. The setting is more general and important.

2. There are some theory insights. Theorem 3.1 claims that exact maximization identifies the interventional Markov equivalence class and the true intervention family under the assumptions.

3. They cover multiple intervention types (shift/scale/noisy-function), multiple noise distributions, unknown targets, and show targeted scaling to 70 nodes. They also provide ablations and runtime comparisons, and evaluate target recovery explicitly

Weaknesses:

1. The approach relies on assumptions that may be restrictive in practice for cyclic systems, especially the equilibrium, invertibility, and contractivity requirements, and the modeling of interventions. It’s unclear how robust SCOUT is when these are violated.

2. Jointly learning graph structure, intervention targets, and flexible nonlinear mechanisms with discrete relaxations is generally nonconvex.

3. A few works are closely related (e.g. [1], [2]). The authors may consider add them to the discussion.

4. The real-data evaluation is interesting, but it uses predictive interventional NLL rather than graph recovery, so it’s hard to conclude SCOUT recovers better structure rather than simply fitting interventions better. Simulators like [3], [4] may be used to generate close to real-world data under given graphical structures to verify.

5. The 70-node scaling is mainly shown for shift interventions. It would be more convincing to see scaling for at least one more intervention type and perhaps more settings. SCOUT is claimed to handle all listed challenges jointly while this is not verified at larger scale.

References:

[1] Kocaoglu, Murat, Amin Jaber, Karthikeyan Shanmugam, and Elias Bareinboim. "Characterization and learning of causal graphs with latent variables from soft interventions." Advances in neural information processing systems 32 (2019).

[2] Jaber, Amin, Murat Kocaoglu, Karthikeyan Shanmugam, and Elias Bareinboim. "Causal discovery from soft interventions with unknown targets: Characterization and learning." Advances in neural information processing systems 33 (2020): 9551-9561.

[3] Dibaeinia, Payam, and Saurabh Sinha. "SERGIO: a single-cell expression simulator guided by gene regulatory networks." Cell systems 11, no. 3 (2020): 252-271.

[4] Li, Hechen, Ziqi Zhang, Michael Squires, Xi Chen, and Xiuwei Zhang. "scMultiSim: simulation of single-cell multi-omics and spatial data guided by gene regulatory networks and cell–cell interactions." Nature methods 22, no. 5 (2025): 982-993.

---

> ### Author Rebuttal · Authors · 2026-03-30
>
> We thank the reviewer for their thorough and constructive feedback. We greatly appreciate their recognition of the importance and generality of causal discovery in the challenging setting of cyclic graphs with nonlinear mechanisms and soft interventions with unknown targets. Below, we address the concerns raised in the review and respond to the reviewer’s questions.
> ## Regarding W1
> We thank the reviewer for this comment. Please refer to our rebuttal answer regarding W1 and W2 of Reviewer vQmt.
> ## Regarding W2
> We thank the reviewer for this comment. We would like to note that non-convexity of the training objective is a more general problem that affects all differentiable causal learning methods [1, 2]. However, empirical evidence suggests that our optimizer can indeed find the global optimum and correctly identify the graph's adjacency and intervention targets in most scenarios, as shown in section 4 (Experiments) in the main paper.
> ## Regarding W3
> We thank the reviewer for this suggestion, we will definitely add these to the discussion in our paper.
> ## Regarding W4
> We thank the reviewer for their comment. We agree that lack of ground-truth graph makes the interpretation of the recovered causal graph more challenging for the Perturb-CITE-seq dataset. As suggested, we use SERGIO [3] to generate close to real-world data under cylic graph structures. We report the graph and intervention target recovery comparison with baselines for 10-nodes and single gene knock-out interventions.
> **Graph recovery**:
> | Method | Mean AUPRC |
> | --- | --- |
> | SCOUT | 0.753 |
> | NODAGS | 0.474 |
> | BACKSHIFT | 0.309 |
> | LLC | 0.309 |
>
> **Intervention target recovery**:
> | Method | Mean AUPRC |
> | --- | --- |
> | SCOUT | 0.978 |
> | BACKSHIFT | 0.432 |
>
> Interpretation: SCOUT outperforms the existing baselines on graph recovery, but cannot achieve perfect recovery because the overall causal mechanism generated by the simulator need not be contractive. Furthermore, SCOUT recovers the intervention targets with near-perfect performance.
> We promise to include the new results presented here in the updated version of the manuscript.
> ## Regarding W5
> We thank the reviewer for raising this concern. We provide another node scaling ablation for "scale interventions" under Gaussian and Gumbel noise settings.
> **Graph Recovery (Mean AUPRC):**
> | Nodes | Gaussian: SCOUT | Gaussian: LLC | Gaussian: BACKSHIFT | Gaussian: NODAGS | Gumbel: SCOUT | Gumbel: LLC | Gumbel: BACKSHIFT | Gumbel: NODAGS |
> | --- | ---: | ---: | ---: | ---: | ---: | ---: | ---: | ---: |
> | 10 | 0.986 | 0.710 | 0.981 | 0.387 | 0.981 | 0.631 | 0.976 | 0.228 |
> | 30 | 0.989 | 0.890 | 0.993 | 0.442 | 0.998 | 0.669 | 0.980 | 0.317 |
> | 50 | 0.978 | 0.937 | 0.985 | 0.652 | 0.999 | 0.739 | 0.965 | 0.254 |
> | 70 | 0.971 | 0.938 | 0.970 | 0.687 | 0.995 | 0.809 | 0.968 | 0.281 |
>
> Both SCOUT and BACKSHIFT perfectly return intervention targets for each setting.
> Interpretation: SCOUT can still learn both graph structure and intervention targets for a large number of nodes in the "scale intervention" setting.
> ## Regarding Q1
> We thank the reviewer for this question. Please refer to our response to Q3 of Reviewer wR6v for a discussion on runtime scaling with graph size.
> ## Regarding Q2
> We thank the reviewer for this insightful question and suggestion. Here we provide BACADI[4] and UT-IGSP[5] baseline comparisons for 10-nodes non-contractive DAG's under Gaussian noise.
> **Graph Recovery (Mean AUPRC):**
> | Mode | SCOUT | BACADI |  UT-IGSP
> | --- | ---: | ---: | ---: |
> | Shift | 0.983 | 0.533 | 0.456 |
> | Scale | 0.996 | 0.669 | 0.728 |
> | Noise | 0.468 | 0.532 | 0.525 |
>
> Except for the Noisy Function/Gaussian setting, SCOUT outperforms both baselines in graph recovery. BACADI likely suffers from limited scalability beyond small graphs, while UT-IGSP is not directly comparable since it outputs an I-Markov equivalence class rather than a single graph; for AUPRC, we used the best graph in that class. For intervention target recovery, SCOUT achieves perfect performance, whereas the baselines remain below perfect. More detailed comparison will be included in the updated version of the manuscript.
> ## Regarding Q3
> We thank the reviewer for this question. To assess SCOUT's insensitivity to random seeds, initialization, and hyperparameters, we conducted additional experiments, which we will include in our updated manuscript.
> ## Regarding Q4
> We thank the reviewer for this insightful question. We will provide the explicit complexity breakdown.
> - [1] Zheng et al., DAGs with NO TEARS: Continuous optimization for structure learning, 2018.
> - [2] Sethuraman et al., NODAGS-Flow: Nonlinear cyclic causal structure learning, 2023.
> - [3] Dibaeinia et al., SERGIO: A single-cell expression simulator guided by gene regulatory networks, 2020.
> - [4] Hagele et al., BacaDI: Bayesian causal discovery with unknown interventions, 2022.
> - [5] Squires et al., Permutation-based causal structure learning with unknown intervention targets, 2020.

---

> > ### Author Rebuttal · Reviewer_yAsG · 2026-04-03
> >
> > I thank the authors for the detailed rebuttal and for adding substantial empirical evidence. I appreciate the new simulator-based evaluation and the additional large-scale results, which strengthen the paper’s empirical case and make me more positive about its practical value. Based on these additions, I am inclined to raise my score from 3 to 4. That said, my concerns are still only partially resolved. In particular, the method continues to rely on restrictive assumptions for cyclic systems, and the rebuttal provides more clarification than convincing evidence about robustness when these assumptions are violated. Some sensitivity, complexity, and modeling questions are also only briefly addressed or deferred to the revision. While the rebuttal clearly improves the paper, the remaining issues are substantive and would benefit from a fuller treatment in the revision, especially with more detailed discussion and analysis of the (semi-)real-world evaluations.

---

> > > ### Author Response · Authors · 2026-04-05
> > >
> > > We thank the reviewer for appreciating our rebuttal. We want to address their further concerns.
> > >
> > > ### Regarding the restrictive assumptions for cyclic systems
> > >
> > > For acyclic structural causal models, unique observational and interventional distributions exist, and the Markov properties hold with respect to $d$-separation without requiring restrictive assumptions. In contrast, for cyclic systems, an SCM must be uniquely solvable with respect to each strongly connected component in order to satisfy the Markov property with respect to $\sigma$-separation [1]. Without guarantees of unique solvability and of the Markov property under $\sigma$-separation, causal discovery is no longer a well-posed problem. For this reason, we restrict both the observational and interventional causal mechanisms in our framework to be contractive; this guarantees unique solvability of both the observational and interventional SCM. At the same time, as demonstrated by our experiments on the (semi-)real-world SERGIO data, our method still empirically outperforms existing cyclic causal discovery approaches even in settings where contractivity is not guaranteed. As suggested by the reviewer, the revised manuscript will include a much more detailed discussion of these points.
> > >
> > > ### Regarding sensitivity, complexity, and modeling questions
> > >
> > > We were not able to include this analysis in our previous rebuttal response due to the character limit. We report these results here.
> > >
> > > ### Regarding Q3
> > >
> > > To measure SCOUT's sensitivity to random seeds and initialization, we performed the following experiment: using the same ground-truth graph and intervention targets with shift interventions under Gaussian noise with a non-linear mechanism, we trained our model 5 times with re-initialization using random seeds. The graph recovery AUPRC has a mean of 0.9854 and a standard deviation of 0.00439; the intervention recovery AUPRC is 1 for each trial. Thus, SCOUT is insensitive to initialization and random seeds.
> > >
> > > To measure the sensitivity of SCOUT for hyperparameters, we tried different parameters for the same graph and targets. The results can be found in the following table:
> > >
> > > | Learning rate $\alpha$ | $\lambda_c$ | $\lambda_r$ | AUPRC | Int. AUPRC |
> > > |---|---:|---:|---:|---:|
> > > | $10^{-2}$ | $10^{-3}$ | $10^{-2}$ | 1.00 | 1.00 |
> > > | $10^{-1}$ | $10^{-3}$ | $10^{-2}$ | 0.95 | 1.00 |
> > > | $10^{-3}$ | $10^{-3}$ | $10^{-2}$ | 1.00 | 1.00 |
> > > | $10^{-2}$ | $10^{-3}$ | $10^{-3}$ | 1.00 | 0.92 |
> > > | $10^{-2}$ | $10^{-3}$ | $10^{-1}$ | 1.00 | 1.00 |
> > > | $10^{-2}$ | $10^{-2}$ | $10^{-2}$ | 0.90 | 1.00 |
> > > | $10^{-2}$ | $10^{-4}$ | $10^{-2}$ | 1.00 | 1.00 |
> > >
> > > Interpretation: SCOUT obtains near-perfect results for a wide range of hyperparameters.
> > >
> > > ### Regarding Q4
> > >
> > > Let $N$ be the number of nodes, $B$ the minibatch size, $M$ the number of samples scored at likelihood time, and $K$ the number of power-series terms in the residual log-det estimator. In the current setup, $\mathbb{E}[K]=4$.
> > >
> > > For one training step, the cost can be written explicitly as
> > >
> > > $$ T_{\mathrm{train}} = \mathcal{O}(BN^{2}) + \mathcal{O}(BN^{3}) + \mathcal{O}(BN^{3}) + \mathcal{O}((K+1)BN^{3}) + \mathcal{O}(BN) + \mathcal{O}((K+1)BN^{3}) + \mathcal{O}(N^{2}). $$
> > >
> > > Here, the terms correspond respectively to sampling the Gumbel adjacency mask, computing $f(x)$, computing $f_i(x)$, the Neumann-series residual log-det term, the 1D spline-flow log-det term, backpropagation, and the optimizer/Lipschitz projection step.
> > >
> > > For one likelihood evaluation on $M$ samples, the cost is
> > >
> > > $$ T_{\mathrm{lik}} = \mathcal{O}(MN^{2}) + \mathcal{O}(MN^{3}) + \mathcal{O}(MN^{3}) + \mathcal{O}(KMN^{3}) + \mathcal{O}(MN). $$
> > >
> > > Here, the terms correspond respectively to sampling the Gumbel adjacency mask, computing $f(x)$, computing $f_i(x)$, the power-series residual log-det term, and the 1D spline-flow log-det term.
> > >
> > > Hence, both training and likelihood evaluation are dominated by cubic scaling in the number of nodes, with overall leading-order costs $\mathcal{O}((K+1)BN^{3})$ and $\mathcal{O}(KMN^{3})$, respectively.
> > >
> > > [1] Bongers et al., *Foundations of Structural Causal Models with Cycles and Latent Variables*, 2021.

---

### Official Review · Reviewer_1WnM · 2026-03-12

**Soundness:** 1
**Presentation:** 1
**Significance:** 2
**Originality:** 2
**Overall Recommendation:** 4
**Confidence:** 4

**Summary:**

This paper focuses on the causal discovery in the cyclic setting due to feedback loops. The concrete problem definition is in unknown-target soft interventions.

While the topic is interesting, the current version seems have **technical confusion** on the definition of 'cyclic' in different literature, like **mixed graph due to latent confounders** and **cyclic SCM due to feedback loops**.

**Compliance With Llm Reviewing Policy:**

Affirmed.

**Final Justification:**

during rebuttal, the author provide important clarification.

**Key Questions For Authors:**

Please see the weakness part

**Limitations:**

Yes

**Strengths And Weaknesses:**

**Strength**



The cyclic causal mechenism due to feedback loops is definitely an important and less explored topic.



**Weakness**

- I acknowledge the importance of the cyclic graphs. However, I believe it would be essential to have a more detailed clarification about the presence of cycles. In section 2.1, the presence of cycles are assumed as a dynamical process under equilibrium conditions. The term 'dynamical' implies the occurrence of time. One common opinion is that some cyclic graphs can be converted to acyclic causal graph with time-lag effect. For example,  $X_t = Y_{t-1} + \epsilon$ and $Y_t = X_{t-1} + \tau$. And the cyclic graph on $X$ and $Y$ may be the marginal observations over a period of time. Therefore, I am worried that the paper implicitly assumes this settings. If so, the formulations and assumptions need further clarification. If not, the author may need to highlight the technical differences of the 'equilibrium' cyclic setting.



- The techniqual issues on applying Lemma A.14 (Lemma A18 in Sethuraman & Fekri, 2025). However, there is a significant gap about how the two paper interpret the cyclic graphs.
  - In the original paper, the cyclic is interpreted as the bidirectional edge in the **mixed graph** due to **latent confounders**
  - While in this paper, the cyclic is interpreted as the **feedback loops** and **assume causal sufficiency with no latent confounders**



- The paper seems also utilize the definition of I-Markov Equivalence for directed mixed graph (Definition A10 in Sethuraman & Fekri, 2025). Correct me if I am wrong. If so, please: (1) add proper citation; (2) address the gap about how the two paper interpret the cyclic graphs (see above).

---

> ### Author Rebuttal · Authors · 2026-03-30
>
> We thank the reviewer for their comments and their acknowledgment for the importance of the cyclic causal discovery. Below, we address the comments and questions raised.
> ## Regarding W1
> We thank the reviewer for their question. The dynamical system we assume is
> $$
> \\mathbf{x}^{(t)} := \\mathbf{f}(\\mathbf{x}^{(t-1)}) + \\boldsymbol{\\varepsilon},
> $$
> where $\\mathbf{x}^{(t)}$ is the random vector collecting all nodes of the system at time $t$, and $\\boldsymbol{\\varepsilon}$ is the noise vector. We assume that $\\mathbf{f}$ is contractive. Therefore, by Banach's fixed-point theorem, for every initial value $\\mathbf{x}^{(0)}$ and every $\\boldsymbol{\\varepsilon}$, the sequence $\\{\\mathbf{x}^{(t)}\\}_{t \\ge 0}$ converges to a unique fixed point $\\mathbf{x}^{\\star}$ satisfying
> $$
> \\mathbf{x}^{\\star} = \\mathbf{f}(\\mathbf{x}^{\\star}) + \\boldsymbol{\\varepsilon}.
> $$
> This is the equilibrium condition assumed in the paper, and uniqueness is guaranteed by the contractivity of $\\mathbf{f}$. Importantly, contractivity ensures convergence to a unique equilibrium, but it does not remove the mutual functional dependence among variables. In particular, different coordinates of $\\mathbf{f}$ may still depend on each other through feedback, so the equilibrium system can still be represented by a cyclic graph.
>
> Thus, the induced structural relations among the equilibrium variables remain cyclic because they are defined through simultaneous fixed-point equations rather than a recursive acyclic factorization. For this reason, the equilibrium model cannot in general be converted into an acyclic causal graph over the same variables simply by noting that the dynamics have converged.
> For example, consider
> $$
> x_1^{(t)} = f_1(x_2^{(t-1)}) + \\varepsilon_1, \\qquad
> x_2^{(t)} = f_2(x_1^{(t-1)}) + \\varepsilon_2.
> $$
> If the joint map is contractive, then the system converges to a unique equilibrium $(x_1^{\\star}, x_2^{\\star})$ satisfying
> $$
> x_1^{\\star} = f_1(x_2^{\\star}) + \\varepsilon_1, \\qquad
> x_2^{\\star} = f_2(x_1^{\\star}) + \\varepsilon_2.
> $$
> The equilibrium is unique and stable, but the dependencies between $x_1^{\\star}$ and $x_2^{\\star}$ are still cyclic.
> ## Regarding W2
> We thank the reviewer for their comment. In [1], cycles are also interpreted as feedback loops among directed edges, rather than as bidirected edges arising from latent confounding. In their paper, the graph is defined as a directed mixed graph $G=(V,E,B)$, where $E$ contains the directed edges encoding causal relations, while $B$ contains the bidirected edges that explicitly represent hidden confounders. They further motivate their setting as one that allows for both feedback loops and hidden confounders, indicating that these are treated as distinct phenomena.
>
> More specifically, their problem setup states that a bidirected edge $i \leftrightarrow j$ indicates the presence of a hidden confounder between $X_i$ and $X_j$, whereas the directed part of the graph captures the causal mechanisms and may be cyclic.
>
> Therefore, [1] considers a more general setting that includes both: (i) feedback loops through directed cycles, and (ii) latent confounding through bidirected edges. In contrast, our paper focuses on the causally sufficient special case, where the first aspect remains present but the second is excluded.
>
> ## Regarding W3
> We thank the reviewer for their suggestion. We will add the necessary citation. The definition of I-Markov equivalence for directed mixed graphs can be specialized to our setting by considering the causally sufficient case, that is, by taking the set of bidirected edges to be empty.
>
> [1] Sethuraman, M. G. and Fekri, F. Differentiable cyclic causal discovery under unmeasured confounders. In The Thirtyninth Annual Conference on Neural Information Processing Systems, 2025.

---

> > ### Author Rebuttal · Reviewer_1WnM · 2026-04-03
> >
> > Thanks for the response. I have re-read the DCCD-CONF paper and the SCOUT paper. I have updated the score to 3.
> >
> > Here is my follow-up questions on W1.
> >
> > Consider such baselines: If we set $\mathbf{U}=\mathbf{x}^{(t)}$, $\mathbf{V}=\mathbf{x}^{(t-1)}$, we can apply acyclic causal discovery methods (those who allows unknown intervention targets) to learn the relation between $\mathbf{U}$ and $\mathbf{V}$. Then one can convert the time-lag acyclic causal graph into the cyclic version.
> >
> > Compared with such baselines, under what situations it would be better to use SCOUT?

---

> > > ### Author Response · Authors · 2026-04-05
> > >
> > > We thank the reviewer for their acknowledgment of our rebuttal and for the follow-up question regarding time-lag effects. In our framework, the data are assumed to arise from equilibrium conditions, so a notion of time step is not well-defined in our model. Accordingly, our datasets are not time series, and SCOUT is not designed to model temporal dependencies across observations. That said, we compared SCOUT against existing acyclic causal discovery frameworks (see Figure 7 and our rebuttal response to Q2 of reviewer yAsG), and SCOUT outperformed these methods in both graph recovery and intervention target recovery.
> > >
> > > We also considered a naive node-wise regression baseline on the equilibrium samples, where each variable $X_i$ is predicted from the remaining variables $X_{-i}$ using Lasso. This produces a predictive graph whose edges correspond to nonzero regression coefficients. In a test setting with shift interventions, where SCOUT achieves perfect graph recovery with an AUPRC of 1.0, this regression-based approach attains an AUPRC of only 0.4789. The reason is that this baseline does not target the same object as SCOUT. Since the data consist of equilibrium samples rather than temporal transitions, the regression captures contemporaneous conditional associations in the equilibrium distribution, rather than direct causal feedback relations. Therefore, while this baseline can recover some dependence signal, it is fundamentally misspecified for directed cyclic graph recovery.

---

### Official Review · Reviewer_wR6v · 2026-03-13

**Soundness:** 2
**Presentation:** 2
**Significance:** 3
**Originality:** 3
**Overall Recommendation:** 3
**Confidence:** 3

**Summary:**

This paper introduces SCOUT, a likelihood-based framework designed to tackle a notoriously tricky intersection in causal discovery: learning nonlinear, cyclic causal structures from soft interventional data where the intervention targets are unknown and the noise is non-Gaussian(i.e the complete opposite of LiNGAM). To achieve this, the authors model the equilibrium Structural Equation Model (SEM) as an invertible map using contractive residual flows, while handling the non-Gaussian noise via neural spline flows. By leveraging sparsity-regularized maximum likelihood, the method jointly learns the graph structure and the intervention targets. The authors provide a large-sample consistency result (under $\sigma$-faithfulness and invertibility assumptions) and evaluate the model on both synthetic data and a real gene perturbation dataset (PerturbCITE-seq), demonstrating solid improvements over several baselines.

**Compliance With Llm Reviewing Policy:**

Affirmed.

**Key Questions For Authors:**

1. **Intervention Parameterization:** The assumption of a single shared intervened mechanism ($\tilde{f}$) across all environments seems highly restrictive for heterogeneous real-world data. Can SCOUT handle environment-specific intervention parameters (e.g., varying shift magnitudes)? If not, how severely does this limit its application?
2. **Ablation Studies:** To justify the architectural complexity, can you provide ablations that isolate the performance impact of (a) non-Gaussian noise modeling via splines versus a fixed Gaussian baseline, and (b) contractivity enforcement?
3. **Computational Complexity:** What are the actual computational costs (wall-clock time, memory footprint) for training as a function of the number of nodes ($d$) and environments ($K$)?

**Limitations:**

No, the limitations are not adequately addressed. The authors need to explicitly discuss the practical fragility of their theoretical assumptions specifically, under what realistic conditions $\sigma$-faithfulness and strict invertibility are likely to be violated in nonlinear cyclic systems. Furthermore, acknowledging the lack of uncertainty quantification in the graph/target predictions would provide necessary context for practitioners looking to use this in scientific pipelines.

**Strengths And Weaknesses:**

**Soundness:**
* **Strengths:** The foundational modeling choices are mathematically sound. Enforcing a Lipschitz constant of $< 1$ via spectral normalization is a standard, reliable way to ensure contractivity and enable unbiased log-det estimation. The use of factorized neural spline flows is an expressive way to model non-Gaussian noise while preserving the independence of exogenous variables.
* **Weaknesses:** The execution suffers from critical practical limitations and missing empirical validations. First, the model assumes a single, shared intervened mechanism ($\tilde{f}$) across all environments. In practice, interventions are often highly heterogeneous, and restricting the model to a shared mechanism is a severe bottleneck for real-world applicability. Second, the theoretical guarantees hinge on strong assumptions like $\sigma$-faithfulness and strict invertibility, with no discussion of how often these hold in messy, cyclic real-world systems. Finally, the complete lack of ablation studies makes it impossible to disentangle *why* the model works. It is unclear if the performance gains stem from the non-Gaussian noise modeling, the contractivity enforcement, or another factor entirely.

**Presentation:**
* **Strengths:** The foundational math is clearly explained, and the motivation for using augmented meta-graphs to connect to Markov properties is well-articulated. The overall narrative flow is logical.
* **Weaknesses:** The paper makes structural choices that actively undermine its main claims. The ability to recover unknown targets is presented as a primary contribution, yet the main text relies on a real-world evaluation using *known* targets, banishing the unknown-target results to the appendix. Furthermore, missing architectural details (e.g., Gumbel-softmax temperature schedules, variance of the log-det estimator) significantly hinder reproducibility.

**Significance:**
* **Strengths:** The problem space is highly relevant. Moving beyond DAGs, linearity, and hard interventions is crucial for domains like perturbational biology.
* **Weaknesses:** While the problem is significant, the restrictive shared-mechanism assumption and the lack of clarity on what drives the model's performance limit the immediate practical impact of the proposed solution.

**Originality:**
* **Strengths:** The work represents a creative and novel synthesis of existing tools. Combining contractive residual flows, neural spline flows, and differentiable structure learning into a unified likelihood framework for this specific, highly relaxed causal setting is a solid contribution.

---

> ### Author Rebuttal · Authors · 2026-03-30
>
> We thank the reviewer for their thoughtful and constructive comments. We appreciate their recognition of the significance of our work on causal discovery with cycles, nonlinearity, and soft interventions, as well as their positive assessment of the paper’s mathematical and modeling foundations. Below, we address their comments and questions.
> ## Regarding W1 of Soundness and Significance and Q1
> We thank the reviewer for pointing out the limitation of a single shared intervention mechanism across different experiments in real-world applications. To address this limitation, we have redefined the modelling of soft interventions in the following way:
> $$
> X_i = f_{i,\\mathrm{int}}^{(k)}(X_{pa_{\\mathcal{G}}(i)}) + \\epsilon_{i,\\mathrm{int}}^{(k)}, \\qquad i \\in I.
> $$
> where the superscript k indicates the k-th intervention. Below we report the results of a synthetic experiments with shift interventions with varying shift magnitudes across different environments. In this case, the model learns a different set of SCM parameters for each interventional experiment.
> ### Graph recovery (AUPRC)
> | Model | Gaussian | Gumbel |
> |---|---:|---:|
> | SCOUT | 0.972 | 0.971 |
> | SCOUT_shared | 0.405 | 0.916 |
> | NODAGS | 0.620 | 0.442 |
> | BACKSHIFT | 0.169 | 0.203 |
> | LLC | 0.412 | 0.395 |
>
> In the tables above, SCOUT\_shared corresponds to the case where the interventional mechanism parameters are shared across the experiments. As the reviewer suggested, SCOUT\_shared suffers from performance degradation in this scenario. After fixing this modeling flaw, we obtained much better results.
> ## Regarding W2 of Soundness
> We thank the reviewer for raising the question of whether our model's assumptions hold in real-world scenarios involving cyclic graphs. What we mean by strict invertibility is actually the existence of a unique equilibrium, for more details please refer to our rebuttal comment regarding W1 and W2 for the Reviewer vQmt.
>
> The faithfulness assumption is generally considered reasonable for DAGs, since, in the case of DAGs, the set of parameterizations of conditional distributions violating the faithfulness assumption has zero Lebesgue measure [1]. For the cyclic case, the generality of $\sigma$-faithfulness remains an open problem. However, $\sigma$-faithfulness remains a standard assumption for several constraint-based methods that handle cyclic graphs [2, 3].
> ## Regarding W3 of Soundness and Q2
> We thank the reviewer for their comments and questions on ablation studies. We would like to clarify that SCOUT doesn't require the endogenous noise to follow any specific class of distributions (Gaussian or non-Gaussian). We train neural spline flows to transform any noise distribution into a standard normal distribution. To measure the impact of this transformation on performance, one may compare results with the NODAGS baseline (which is designed for learning non-linear cyclic graphs under the Gaussian noise assumption). Regarding the effect of contractivity assumption, please refer to our rebuttal comment regarding W1 and W2 for the Reviewer vQmt.
>
> In appendix B, we provide a sensitivity analysis with respect to the parameters of the data generating process. We will include additional results for cycle and non-linearity sensitivity in the updated version of the manuscript.
>
> ## Regarding W1 of Presentation
> We thank the reviewer for this thoughtful comment. Since the dataset provides ground-truth intervention targets, we showcase the known-targets setting in the main paper as it yields the best performance for our model. The unknown-targets results are included in the appendix for completeness and show only a modest drop in performance, demonstrating that our model remains competitive even without target information. We have additionally conducted experiments on SERGIO in response to comment W4 for Reviewer yAsG, where the true graph is known.
> ## Regarding W2 of Presentation
> We thank the reviewer for this insightful comment. We will add complete details about the model architecture in the revised manuscript.
> ## Regarding Q3
> We thank the reviewer for this question. We have computed the per-epoch computation time as well as training memory footprint for SCOUT as a function of nodes and experiments.
> | Nodes | Experiments | Time (s) | Mem Est. (KiB) |
> |---|---:|---:|---:|
> | 10 | 10 | 0.59 | 21.77 |
> | 30 | 30 | 8.3 | 102.7 |
> | 50 | 50 | 75 | 233.6 |
> | 70 | 70 | 197 | 414.6 |
> | 10 | 20 | 1.20 | 23.33 |
> | 10 | 30 | 1.80 | 24.89 |
>
> These results indicate that the model scales approximately linearly with the number of experiments, whereas the main computational bottleneck is scaling with the number of nodes.
>
> - [1] Meek, Strong completeness and faithfulness in Bayesian networks, 2013.
> - [2] Forré et al., Constraint-based causal discovery for non-linear structural causal models with cycles and latent confounders, 2018.
> - [3] Mooij et al., Constraint-based causal discovery using partial ancestral graphs in the presence of cycles, 2020.

---

> > ### Author Rebuttal · Reviewer_wR6v · 2026-04-03
> >
> > I thank the authors for the response. However, my concerns are not fully addressed.
> > (W1): The new results for the non-shared mechanism ($f_j^{(k)}$) are encouraging and show a significant performance boost. This addresses my main concern regarding real-world applicability. I would like to see how the authors plan to integrate this into the final manuscript—specifically, does this change significantly increase the number of parameters to be learned, and how does it affect the risk of overfitting in low-sample environments?
> > (W3): While comparing against NODAGS provides some context for the Gaussian assumption, it is not a substitute for an internal ablation. To truly understand the contribution of the Neural Spline Flows (NSF), it would be more insightful to see a version of SCOUT where the NSF is replaced by a simple Gaussian likelihood. This would disentangle whether the gains come from the flow-based noise modeling or the contractive residual structure itself.

---

> > > ### Author Response · Authors · 2026-04-05
> > >
> > > We thank the reviewer for their follow-up questions regarding the non-shared mechanism and the role of the Neural Spline Flows.
> > >
> > > ### Regarding W1
> > >
> > > We ran an additional ablation comparing SCOUT and SCOUT\_shared while varying the number of samples. First, we used the same setting described in our rebuttal, where the shift magnitudes vary.
> > >
> > > | Samples | Model | Mean AUPRC | Training Time per Epoch (s) |
> > > |--------:|:------|-----------:|----------------------------:|
> > > | 100 | SCOUT | 0.8307 | 0.25 |
> > > | 100 | SCOUT\_shared | 0.7668 | 0.06 |
> > > | 250 | SCOUT | 0.9413 | 0.60 |
> > > | 250 | SCOUT\_shared | 0.7488 | 0.15 |
> > > | 500 | SCOUT | 0.9908 | 1.20 |
> > > | 500 | SCOUT\_shared | 0.4709 | 0.30 |
> > >
> > > We also ran the models in a setting with the same shift magnitudes.
> > >
> > > | Samples | Model | Mean AUPRC |
> > > |--------:|:------|-----------:|
> > > | 100 | SCOUT | 0.8637 |
> > > | 100 | SCOUT\_shared | 0.8916 |
> > > | 250 | SCOUT | 0.9481 |
> > > | 250 | SCOUT\_shared | 0.9194 |
> > > | 500 | SCOUT | 0.9788 |
> > > | 500 | SCOUT\_shared | 0.8478 |
> > >
> > > From the tables above, we observe that SCOUT generally achieves better results than SCOUT\_shared, while not showing evidence of overfitting in the low-sample settings we tested. However, as the reviewer points out, this comes at the cost of an increased number of parameters to be learned, and therefore increased training time. Since we learn a different mechanism for each node in each environment, the additional complexity increases approximately linearly with the number of environments and the number of nodes.
> > >
> > > In the final manuscript, we will integrate the non-shared mechanism into the discussion of the main problem setup for modeling interventions, and we will also include the experiment presented in our rebuttal. In addition, we will add these tables as further ablations in the appendix. At the same time, we will keep SCOUT\_shared as an optional mode of the model for settings in which shared mechanisms across interventions are a reasonable assumption, due to its computational efficiency.
> > >
> > > ### Regarding W3
> > >
> > > We thank the reviewer for this insightful suggestion. We conducted an additional ablation comparing SCOUT with a variant of SCOUT in which the Neural Spline Flows (NSFs) are replaced by a simple Gaussian likelihood. The resulting graph recovery and intervention target recovery performances (mean AUPRC) for the unknown-target and known-target settings are reported in the tables below.
> > >
> > > **Unknown targets**
> > >
> > > **Graph recovery:**
> > >
> > > | Mode | Gaussian SCOUT | Gaussian SCOUT\_without\_nsf | Exponential SCOUT | Exponential SCOUT\_without\_nsf | Gumbel SCOUT | Gumbel SCOUT\_without\_nsf |
> > > |:-----|---------------:|-----------------------------:|------------------:|--------------------------------:|-------------:|---------------------------:|
> > > | Shift | 0.936 | 0.151 | 0.985 | 0.287 | 0.941 | 0.298 |
> > > | Scale | 0.953 | 0.406 | 0.953 | 0.329 | 0.963 | 0.408 |
> > > | Noise | 0.522 | 0.520 | 0.969 | 0.484 | 0.935 | 0.516 |
> > >
> > > **Intervention target recovery:**
> > >
> > > | Mode | Gaussian SCOUT | Gaussian SCOUT\_without\_nsf | Exponential SCOUT | Exponential SCOUT\_without\_nsf | Gumbel SCOUT | Gumbel SCOUT\_without\_nsf |
> > > |:-----|---------------:|-----------------------------:|------------------:|--------------------------------:|-------------:|---------------------------:|
> > > | Shift | 1.000 | 0.144 | 1.000 | 0.471 | 1.000 | 0.237 |
> > > | Scale | 1.000 | 0.227 | 0.975 | 0.232 | 1.000 | 0.171 |
> > > | Noise | 0.264 | 0.393 | 0.590 | 0.443 | 0.607 | 0.463 |
> > >
> > > **Known targets**
> > >
> > > **Graph recovery:**
> > >
> > > | Mode | Gaussian SCOUT | Gaussian SCOUT\_without\_nsf | Exponential SCOUT | Exponential SCOUT\_without\_nsf | Gumbel SCOUT | Gumbel SCOUT\_without\_nsf |
> > > |:-----|---------------:|-----------------------------:|------------------:|--------------------------------:|-------------:|---------------------------:|
> > > | Shift | 1.000 | 0.999 | 0.984 | 0.973 | 0.991 | 0.984 |
> > > | Scale | 0.982 | 0.985 | 0.972 | 0.712 | 0.987 | 0.976 |
> > > | Noise | 0.974 | 0.992 | 0.982 | 0.671 | 0.991 | 0.956 |
> > >
> > > From these results, we conclude that without NSF, the intervention targets are unidentifiable and graph recovery is not possible in the unknown-target setting. In contrast, when the intervention targets are known, if the noise is Gaussian, the contractive residual structure alone is sufficient for graph recovery. We will definitely add these results to the ablation study in the appendix.

---

### Decision · Program_Chairs · 2026-04-30

**Decision:**

Accept (regular)

**Comment:**

Reviewers agreed that this submission is well motivated, clear, and mathematically sound; tackles an important and under-explored problem; and does so by combining existing tools in an original manner.

While reviewers also initially raised concerns regarding the lack of certain ablations and experimental comparisons, real world applicability, and the modelling of interventions via a single shared mechanism, these have been largely addressed by the authors during the rebuttal process.

Povided that the additional experimental results, extension of the method to multiple interventional mechanisms, and additional explanations and discussions are included in the revised manuscript, this will be a solid contribution to the literature on cyclic causal discovery.